# Laser-driven x-ray and proton micro-source and application to simultaneous single-shot bi-modal radiographic imaging

T. M. Ostermayr [1,2,3✉], C. Kreuzer[1], F. S. Englbrecht[1], J. Gebhard[1], J. Hartmann[1], A. Huebl [3], D. Haffa[1], P. Hilz[1,5], K. Parodi[1], J. Wenz[1], M. E. Donovan[4], G. Dyer [4], E. Gaul[4], J. Gordon[4], M. Martinez[4], E. Mccary [4], M. Spinks[4], G. Tiwari [4], B. M. Hegelich[4] & J. Schreiber [1,2✉]

Radiographic imaging with x-rays and protons is an omnipresent tool in basic research and applications in industry, material science and medical diagnostics. The information contained in both modalities can often be valuable in principle, but difficult to access simultaneously. Laser-driven solid-density plasma-sources deliver both kinds of radiation, but mostly single modalities have been explored for applications. Their potential for bi-modal radiographic imaging has never been fully realized, due to problems in generating appropriate sources and separating image modalities. Here, we report on the generation of proton and x-ray micro-sources in laser-plasma interactions of the focused Texas Petawatt laser with solid-density, micrometer-sized tungsten needles. We apply them for bi-modal radiographic imaging of biological and technological objects in a single laser shot. Thereby, advantages of laser-driven sources could be enriched beyond their small footprint by embracing their additional unique properties, including the spectral bandwidth, small source size and multi-mode emission.

[1] Ludwig-Maximilians-Universität München, Fakultät für Physik, 85748 Garching, Germany. [2] Max-Planck-Institut für Quantenoptik, 85748 Garching, Germany. [3] Lawrence Berkeley National Laboratory, Berkeley, CA 94720, USA. [4] Center for High Energy Density Science, University of Texas at Austin, Austin, TX 78712, USA. [5] Present address: Helmholtz Institute Jena, 07743 Jena, Germany. ✉email: tmostermayr@lbl.gov; joerg.schreiber@lmu.de

In the second half of the twentieth century, imaging methods based on ions[1–4] and x-rays[5,6] have become increasingly powerful and today influence many aspects of modern life. They can provide complementary information, e.g., different sensitivity and resolution for low and high-density features. However, most conventional sources of ions and x-rays currently provide only a single species of radiation. Hence, the most common method to combine information from both has been to use complex algorithms, registering images obtained in completely separate image acquisition processes on top of each other[7,8]. Different acquisition geometries for both modalities, or dynamic objects, e.g., living organisms[9] or plasma-instabilities[10], represent natural challenges that could lead to imperfections in the resulting combined images. In parallel to advances in imaging, soon after the construction of the first laser[11], laser-driven particle[12], and secondary light sources[13] have been studied. The adoption of chirped pulse amplification[14] for optical laser pulses[15] increased available laser peak intensities to a regime where targets are ionized easily, electron velocities become highly relativistic, and secondary processes like ion acceleration or x-ray generation at high energies can become effective and potentially useful[16,17].

Laser-plasma sources have been considered and explored for many classical single-source applications, mostly for their promise to provide a more compact source based on higher applicable field strengths, e.g., refs. [18–23]. Earlier work[24,25] proposed the interesting perspective to use both the x-rays and protons driven in a laser-foil interaction in a simultaneous radiographic imaging with few-µm resolution. However, the approach faced limitations in terms of real applications: the limited proton kinetic energy up to 2.5 MeV and limited spectral modulation in the proton beam only enabled basic binary imaging of a binary mesh structure. An increase in proton energy via higher laser energies seemed feasible, but came with conceptual problems. In laser-foil interactions, energetic electrons are typically emitted in a similar angular range as the protons, and with increasing laser energy they would gain higher kinetic energies as well. Thus, electrons would increasingly contaminate the x-ray image, which was recorded directly behind the proton detector on an imaging plate that is sensitive to all kinds of ionizing radiation. More recently, it has been found that with PW-class lasers, the structure of the transmitted laser beam can also imprint onto the proton beam profile[26]. Similarly, laser-driven plasma-instabilities can lead to non-uniform proton beam profiles[10,27]. Both effects can make radiography in the laser propagation direction ambiguous. In addition, with more laser energy being focused to the same spot size as a lower energy laser, the target-foil-area, which is driven by sufficient intensity to emit protons and x-rays is naturally increased. Similarly, the divergence of hot electrons generated in the laser-plasma interaction increases with laser intensity[28]. These effects would result in a larger source size and reduced image resolution. Although a small virtual source for protons can often be maintained due to their laminar flow[29], the x-ray spot size depends on this effect[30].

A laser-driven source based on a micro-needle-target solves these issues in a combined approach and optimizes the emission characteristics for advanced imaging. The spectral characteristics of protons emitted towards the side is found to be peaked around 10 MeV, featuring a 20%-level energy spread (full width at half maximum (FWHM)). The protons stem from a nm-thin CH contaminant layer present on the tungsten surface; the limited spatial extent translates to a limited spectral bandwidth observed toward the side, as the protons explode away from the positively charged target into vacuum. Spectra with a limited bandwidth (as opposed to monoenergetic or ultrabroad spectra) can benefit the purpose of proton radiographic imaging with a suitable detector

that is sensitive in a specific spectral range (beyond the spectral peak). Then, a measured proton count corresponds uniquely to a specific object thickness, because only a specific part of the proton spectrum is transmitted through the object in such a way, that its residual energy corresponds to the detector's sensitive bandwidth. In this configuration, the proton's initial spectral modulation depth translates directly into achievable contrast (neglecting other factors like detector noise), whereas the spectral width determines the range of object thicknesses that can be imaged. These considerations have been a challenge for early ion-radiographic imaging[2] based on conventional accelerators delivering highly monoenergetic (% level) beams. These could produce almost binary radiographic contrast, but only under perfectly matched conditions for the imaged object. On the other hand, a perfectly flat spectrum would not produce radiographic contrast, because each sample thickness would be represented by the same particle count in the detector. Contrasting these two extreme scenarios, spectral distributions as measured here feature some, but not an infinite, spectral bandwidth. Hence, they allow proton radiographic imaging in a range of object thicknesses with reasonable contrast without additional preparations of the object, beam or detector. For example, a proton spectrum that decays linearly from its maximum at 10 MeV towards zero count at 15 MeV, and a detector that is sensitive between 10 and 11 MeV, will produce contrast for objects in a water equivalent thickness range from 0 mm to ~1 mm.

The emission geometry of protons and x-rays from the target enables image recording with both modalities towards the side (i.e., at 90° with respect to the laser propagation): high-energy electrons (10 MeV-range) and the transmitted laser pulse propagate mostly along the laser propagation direction and are thus spatially separated from the imaging beams. They do not harm the nearby objects or contribute to the x-ray image, which is, again, detected on an imaging plate with sensitivity to all ionizing radiation. As an additional measure, a magnetic x-ray cleaner in front of the x-ray detector can prevent residual electrons and ions from contributing to the image. The source can maintain a small size despite the higher laser energy, due to the confinement of fast electron transport, and therefore x-ray generation, to the small volume of the needle target. Two effects can further reduce this volume: the curved target surface can lead to a converging plasma movement in the laser interaction, and the plasma expansion into vacuum can reduce the high-density region effectively by losing part of the plasma to lower density regions.

Here, we demonstrate these source capabilities in proof-of-principle experiments and record bi-modal x-ray and proton radiographies of biological and technological objects in individual laser shots. This implementation could be easily adapted for applications that previously used single species radiography successfully[31,32]. The technique also promises a viable approach for simultaneous bi-modal imaging with future multi-PW lasers, adding potential applications as x-ray and proton energies keep rising. This research aims to increase the scope and bandwidth of potential applications for laser-plasma accelerators, which benefit from more than just their compact size.

## Results

**Source characterization.** In the first step, we characterize the multi-modal radiation field, which is emitted from a laser-irradiated micro-target with the setup depicted in Fig. 1a. We use the Texas Petawatt laser, focused by an f/2.5 off axis parabolic mirror via a single in-line plasma mirror onto commercial tungsten needle targets, such as used in scanning tunneling microscopes. The interaction point is set to a position where the laser focus and needle diameters are matched at 5 µm.

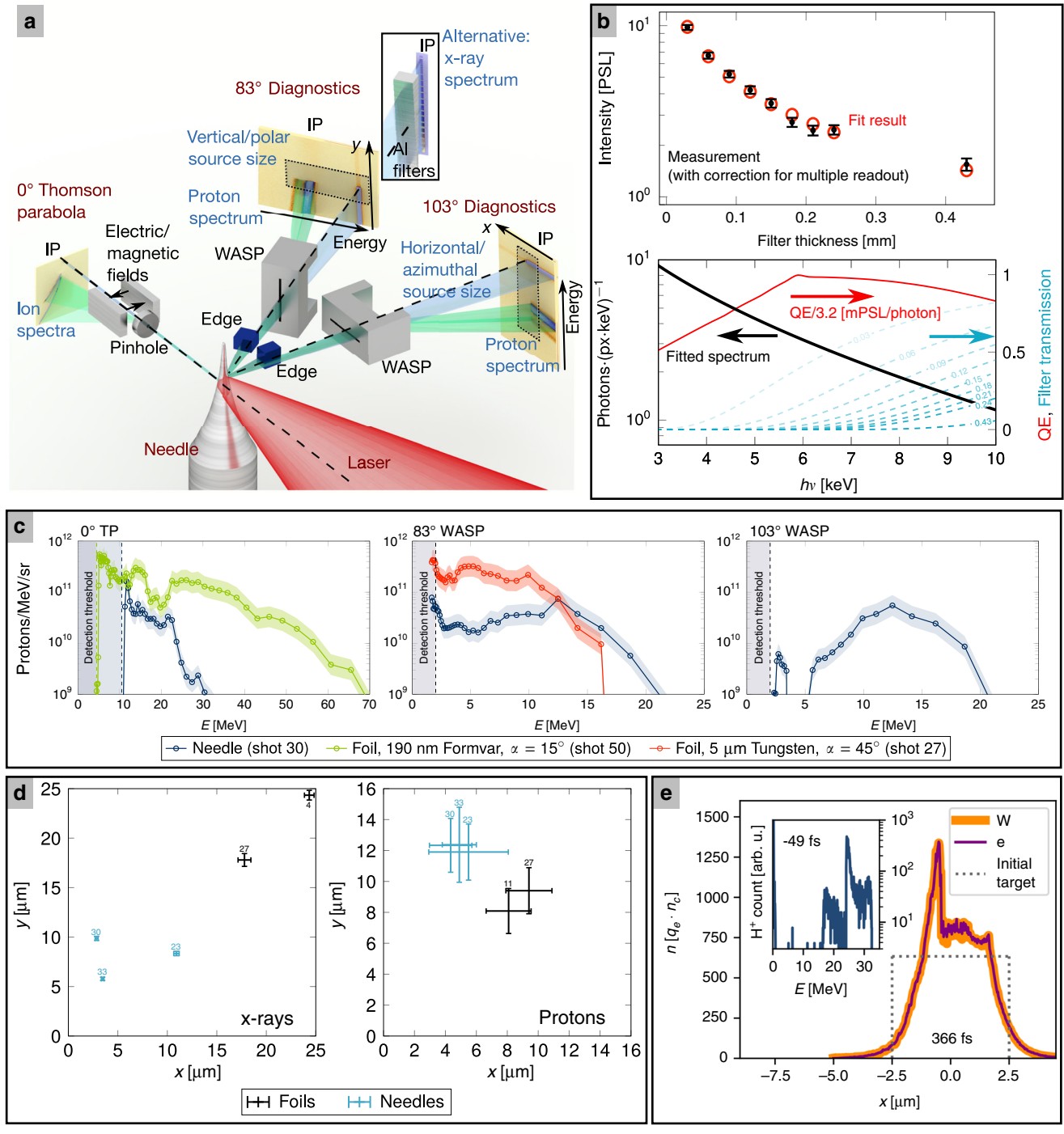

At 83° within the horizontal plane, a wide-angle spectrometer (WASP) with vertically aligned steel slit cuts a fan-beam out of the multi-modal radiation field and magnetically deflects charged particles away from the direct slit projection. Aluminum filters of varying thickness cover the non-deflected x-ray beam (blue). The experimental data (Fig. 1b), the known detector quantum efficiency and filter transmissions can be used to fit a spectral distribution of the form $dN/dE(E) = N_0 \cdot E^{-1} \cdot \exp(-E/k_B T_e)$, corresponding to Bremsstrahlung from a plasma[33] with electron temperature $k_B T_e$. Here, $E = h\nu$ is the photon energy and $N_0$ is a normalization factor. With a fit temperature of $k_B T_e = 8060 \pm 940$ eV (95% confidence level) and considering emission in the full solid angle, this spectrum

contains an energy of 1.6 mJ, of which >1.2 mJ (>$10^{12}$ photons) are emitted at energies higher than 2 keV. It shall be mentioned that recent experiments at comparable conditions using tungsten wire targets[34,35] found similar spectra, with high-resolution HOPG spectrometers additionally identifying broadened emission structures around 8.2–8.4 keV from electron transitions into the L-shell of highly ionized tungsten. However, the Bremsstrahlung content was identified as dominant, consistent with our fit. Note that our detector setup is laid out for these relatively low-energy x-rays. Higher energy MeV-scale x-rays may also be emitted from the laser-plasma interaction, but are not expected to significantly contribute to the recorded signal.

**Fig. 1 Source characterization. a** Setup for recording x-ray and proton spectral information and effective source sizes in horizontal/azimuthal and vertical/polar directions. Green/blue beams indicate ions/x-rays, respectively. Laser, target, and detector systems are labeled in red, detector sub-systems and coordinate-systems in black, measured quantities in blue. Wide-angle spectrometers (WASP, see text) for proton spectral distribution combine with edges for source size measurements. All data are recorded on Fuji BAS-TR imaging plates (IP). **b** (Top) X-ray transmission measurement through different thicknesses of Aluminum filters in terms of photo-stimulated luminescence (PSL); measured data in black (error bars show the SD), fit result in red. (Bottom) Fitted x-ray spectrum in photons per pixel and keV, detector quantum efficiency (QE) in mPSL per photon and filter transmissions for the different filters (thickness specified in mm). **c** (Left) Proton spectra recorded for needle targets and foil reference shots in the laser propagation direction, (Center) at 83° in the horizontal plane and (Right) at 103°. The point distance represents the spectrometer resolution, which is limited by the slit-/pinhole-width for WASP/Thomson parabola, respectively. Error bands estimate absolute accuracy including detector calibration. Shot-to-shot fluctuations displayed in supplementary note 1. **d** Effective source size for x-rays and protons, measured in foil- and needle-shots. Numbers refer to the campaign shot number. The effective source size for tungsten foils was only measured in y direction (polar/vertical) due to the narrow emission geometry and assumed to be similar for the orthogonal (x) dimension. Error bars denote the 95% confidence level from the fit. **e** Particle-in-cell simulation (cf. methods). Proton spectrum observed towards the side shows similarity to experimental data (strongly peaked). The density lineout along the laser propagation (i.e., representing the source size towards the side) indicates how the measured x-ray source size in horizontal direction can appear smaller than the initial target size after hole boring and expansion have reduced the bulk target size. The laser travels from left to right. Figures **a**–**d** adapted with permission from "Relativistically Intense Laser Microplasma Interactions" by Tobias Ostermayr, Springer Thesis, (2019).

In the magnetic field of the WASP, protons (green) are deflected according to their energy, allowing for the reconstruction of their spectral distribution. Figure 1c shows data of the WASP at 83°, an additional WASP at 103° using a horizontally aligned slit, and a Thomson parabola (TP) at 0°. As a reference for proton spectra obtained from needle targets, we use laser-shots onto two different kinds of foil targets featuring cm-scale transverse size; 5 μm thick tungsten foil at 45° and 190 nm thin plastics (Formvar) foil at 15° angle between the target surface normal and the laser propagation direction. Along the laser propagation direction (0°), needles produce a spectrum that decays towards the maximum energy of 20–30 MeV. Both reference shots on foils show the familiar broadband spectra; their emission angle is limited around the target normal and the laser propagation direction, respectively. For the thin-foil target, the maximum proton kinetic energy exceeds 60 MeV at much reduced particle counts around ~$10^9$ protons/(MeVsr).

Towards both sideport spectrometers, we consistently measure peaked proton energy distributions from needle targets. The bandwidth of the peak is as small as 2–5 MeV (FWHM) with particle counts exceeding $10^{11}$ protons/(MeVsr) for several shots, corresponding to >$10^{11}$ protons/sr in the peak (cf. linear plot in supplementary note 1). Although these distributions fluctuate in peak energy from 7 to 20 MeV and particle count from $10^{10}$–$2.5 \times 10^{11}$ protons/(MeVsr) on a shot-to-shot basis owing to sensitivity to focus-pointing jitter, both measured sideport spectra correlate surprisingly well (see supplementary note 1), which principally enables simultaneous measurements of multiple radiographic images and corresponding input proton energy distributions at different angles within the horizontal plane. 2D3V particle-in-cell simulations (Fig. 1e, details in methods and supplementary note 4) confirm the qualitative shape of this spectral distribution; note that the 2D geometry is known to overestimate the absolute energy scale. A proton source with a peaked energy spectrum beyond 10 MeV has long been desired from compact laser-driven accelerators, because it would boost their relevance for many applications originally developed with conventional accelerators, which per default emit a very narrow proton energy spectrum. Possibly, the most impactful application for a tightly peaked proton spectrum, is the radiotherapy of cancer, where the sharp dose deposition at the Bragg peak is used for targeted dose delivery to tumors. Significant progress toward peaked spectra in the 10 MeV range have only been reported in recent efforts exploring advanced acceleration mechanisms[36,37]. In the present setup, the formation of a peaked spectrum is

facilitated by a combination of space-charge effects between different ion species (highly ionized tungsten ions and protons) and the localization of protons in a very thin (nm-scale) contamination layer on a needle-target that does not exceed the focus in size; both effects are known to facilitate quasi-monoenergetic ion spectra[38–40].

In order to evaluate the effective source size for protons and x-rays, we use the image blur of knife edges (Fig. 1a and supplementary note 2), represented in the experiment by sharp silicon edges of few cm effective thickness. The effective x-ray source size extracted from the measured edge spread functions is displayed in Fig. 1d. The effective source size reflects the symmetry of the needle-target with the larger 6–10 μm extent measured along the needle (y axis) and down to 2.8 μm along the horizontal plane (x axis). Note that measurements in the horizontal direction are close to the lower detection limit (~2 μm), which could imply even smaller virtual source distributions. In comparison with foil-shots, the needle target shows a reduced source size. This can be pivotal in applications using the x-ray in-line phase contrast rather than the attenuation for imaging[41]. Here this was achieved by use of a special target. The fact that measured source distributions along the horizontal direction (perpendicular to the needle axis) are even smaller than the needle itself, may be attributed to several factors. First, in the horizontal, laser radiation pressure can induce a converging plasma movement from the target surface to the center. This motion occurs roughly at the hole-boring velocity $v_h = 2a_0 c (Z/A \cdot m_e/m_p \cdot n_c/n_e)^{0.5}$, where $Z$ is the average charge state of the ions, $A$ is the ion mass number, $m_e$ is the electron mass, $m_p$ is the proton mass, $n_c$ is the critical density and $n_e$ is the electron density. Integration of $v_h$ from before the pulse up to the peak intensity estimates a movement of the initial target surface between 250 nm ($Z = 10$, $n_e/n_c = 300$) and 900 nm ($Z = 60$, $n_e/n_c = 100$). While this hole-boring motion is driven directly by the laser, another factor is the plasma expansion into vacuum, which also occurs at sides not facing the laser directly. This expansion reduces the density at the initial target-vacuum interface, and effectively leaves a reduced size of the high-density plasma region, where x-ray generation via Bremsstrahlung is strongest. The resulting overall reduction in bulk target size is usually observed when a high-power laser interacts with a curved surface, e.g., simulations in references[42,43]. Here, high-resolution 2D3V particle-in-cell simulations (Fig. 1e, methods and supplementary note 4) have been performed; they show that the 5 μm target is transformed to a 3 μm plateau structure close to the

original density when looked at through a central lineout along the laser propagation direction, i.e., representing the source distribution toward the sideport. The laser-irradiated side of this plateau structure shows an even smaller density-enhanced region owing to the hole boring into the target. Meanwhile, the density around the plateau decreases exponentially. Further studies will elucidate the origin and temporal structure of the x-ray source to optimize this potentially useful feature.

The effective source size of protons also reflects the symmetry of the target, with larger effective source size of 12 µm in the direction of the needle axis ($y$ axis, vertical). This source size exceeds those measured for planar Tungsten foil-targets. The source size measured orthogonal to the needle axis ($x$ axis, horizontal plane) is in the 5-µm-range, matching the target size. Resolving these subtle differences in source size validates the employed measurement method. Overall the proton source size does not exceed 20 µm and is therefore not expected to significantly blur images. At particle counts reaching $10^8$ protons/cm$^2$ at 0.5 m distance, Columbia Resin #39 (CR39) detectors operate close to their point of saturation assuming 1 µm diameter for particle tracks. As an intermediary conclusion both the proton and the x-ray source are capable of producing radiographic contrast images at reasonable source-sample and sample-detector distances given their measured characteristics.

**Bi-modal imaging.** In the second step, we combine both modalities to record two radiographic images simultaneously within a single laser-shot using the setup depicted in Fig. 2a. The edges for source size measurements are removed. The proton image is recorded on a CR39 detector positioned directly before the WASP

detector at 103° in the horizontal plane at 0.5 m distance from the source. In this experiment, sample objects are mounted directly on top of the CR39 detector. In general, the broad spatial distributions of x-ray and proton sources allow us to place sample objects within the horizontal plane, fairly close (cm-scale) to the interaction, yet outside of the incident and transmitted laser beam to omit optical or physical sample-damage.

The WASP at 103° was modified by increasing its slit-diameter to 2 cm. The updated device was renamed according to its purpose as an x-ray cleaner. It magnetically deflects potential residual charged particles behind the CR39 detector away from the direct projection of the widened slit onto the x-ray detector. This projection is used to record the x-ray image with an IP at 0.75 m distance from the proton image at a magnification factor of 2.56. We used CR39 directly in front of the x-ray detector in separate laser shots, to verify that no significant amount of neutral particles (neutrons or neutral atoms) contaminate the x-ray field of view.

An example of a recorded image of a biological sample is depicted in Fig. 2b showing strong contrast. The difference in image resolution between x-rays and ions is easily observed. Resolution in proton images is always reduced owing to multiple Coulomb scattering[44,45] in the imaged object and in the detector. Meanwhile the crisp x-ray image in the current setup is limited by the detector resolution and magnification. This difference is highlighted in Fig. 2c, where the x-ray lineout resolves features of ~100 µm width, whereas the proton image resolves on the 0.5-mm-scale only.

Increasing magnification for the x-ray image in the point-projection via reduction of the sample distance from the source to

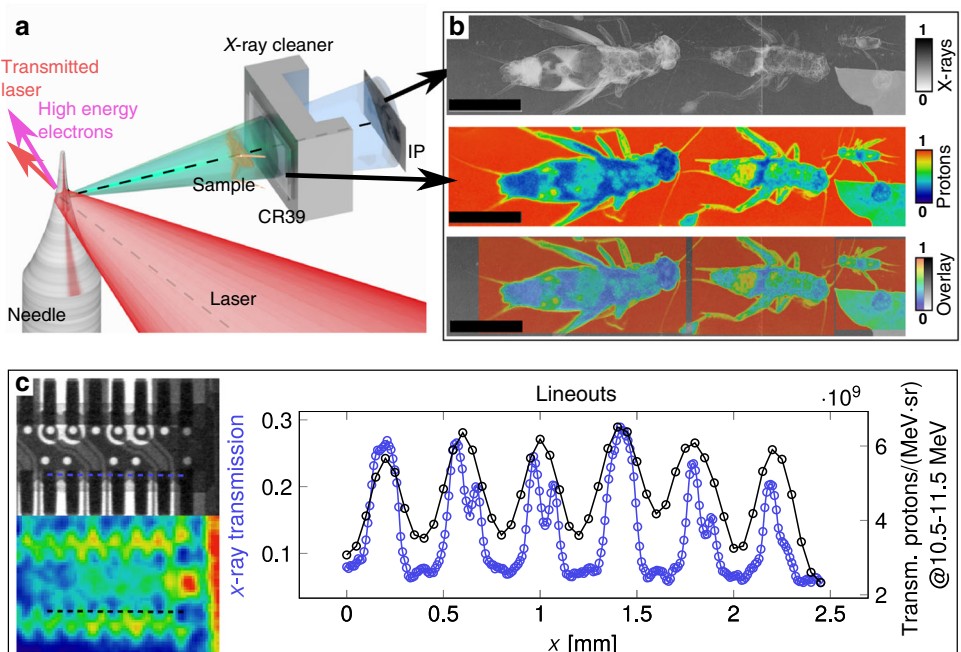

**Fig. 2 Bi-modal imaging. a** Schematic of the setup for bi-modal imaging. Ions are indicated in green, x-rays are indicated in blue. The sample is placed on a CR39 detector, which registers the proton image. Behind the CR39 and x-ray cleaner, an IP records the cleaned x-ray projection. **b** (Top) X-ray image of house crickets (acheta domestica, varying age/size) recorded in a single laser-shot. (Center) Proton image of crickets, recorded on CR39 in the same laser shot. The image was processed and recorded with a technique adapted from references[68,69] and records ion-impacts on the front (1.6–5 MeV protons) and back (10.5–11.5 MeV protons) surfaces. (Bottom) Overlay of both images, with the proton image scaled to 60% opacity. Scale bars correspond to 10 mm. **c** X-ray (Left, top) and proton (Left, bottom) radiographies of a technical sample (part of a smartphone camera). Here, the proton image on the backside of CR39 was recorded with a microscope, counting single proton impacts. The plot shows the resulting histogram with 50 µm pixel size and a smooth filter that replaces each pixel with the average of its 3 × 3 neighborhood. (Right) Lineouts, as indicated by dashed lines in both radiographies within the images. Adapted with permission from "Relativistically Intense Laser Microplasma Interactions" by Tobias Ostermayr, Springer Thesis, (2019).

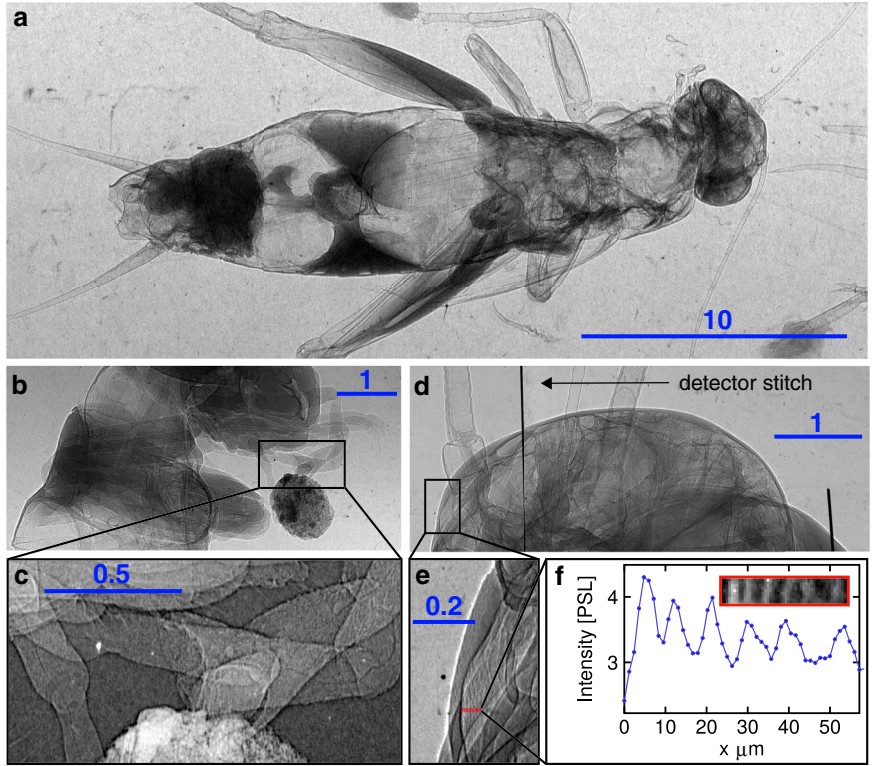

**Fig. 3 Phase contrast enhanced imaging. a** Single shot 2.56-fold magnified absorption image of a cricket (acheta domesticus) that is 2.5 cm long in total. No phase contrast owing to low magnification. **b** Single shot 12.3-fold magnified image of cricket showing edge enhancement owing to phase contrast. **c** Zoomed section from **b**. **d** Single shot 21-fold magnified image of a cricket head. **e** zoomed section of **d**. **f** Lineout (red) taken from **d**, inset showing the zoomed section. All scale bars are in units of mm. Adapted with permission from "Relativistically Intense Laser Microplasma Interactions" by Tobias Ostermayr, Springer Thesis, (2019).

few cm (while leaving source-detector distance constant) reaffirms the effective x-ray source size measured earlier, by resolving details down to the sub-10-µm level and revealing the expected edge enhancements via phase-contrast, as shown in Fig. 3.

## Discussion

In conclusion, a laser-driven x-ray and proton micro-source has been presented and characterized with respect to its key properties. We want to briefly summarize these findings and put them in context with other laser-driven single species sources, before discussing perspectives in bi-modal imaging.

The x-ray source size in the few-µm-range and the photon fluence facilitate single-shot high-resolution radiographies of biological and technological samples. Edge enhancements in the x-ray image appear owing to phase contrast and together with the resolution of sub-10-µm features confirm source size measurements. The broad angular distribution enables radiographies of multiple (and comparably large) samples at once. The x-ray spectrum assumed here is broad; appropriate filtering could enable other spectral shapes. Alternatively, different target materials could facilitate the emission of quasi-monochromatic x-rays, e.g., via K-alpha or He-alpha emission of highly ionized Copper at similar bulk temperatures. Several other laser-driven x-ray sources enable propagation based phase contrast imaging and shall be mentioned here for comparison. Among those are betatron radiation of laser plasma accelerators[21], Thomson scattering[46] and K-alpha emission from solid targets[47,48]. Betatron radiation is per default a broadband source with small source sizes (~µm), and small divergence angles (~mrad). This usually

requires multi-shot stitching of images in order to capture objects of a meaningful size (e.g., few mm)[49] or large distances. Thomson scattering using laser wakefield accelerators can produce quasi-monochromatic photon beams, with a small source size at potentially higher photon energies. Such sources also feature inherently small divergence angles (~$1/\gamma_e$) and typically less total photons per shot. At last, K-alpha sources feature larger divergence angles (similar to the source presented here), can be very narrow band and have a small source size as long as powered by low intensity lasers. In such regimes they typically require many shots to produce sufficient signal to noise ratios in images[47]. Larger photon numbers result from higher laser intensity, but advanced target geometries (e.g., needles) are required to maintain a small source size and spectral lines broaden[47,48]. These comparisons show that the presented x-ray source establishes a quite favorable regime of operation with divergence angle, source size, fluence, and photon energy well suited for the single-shot imaging (and phase contrast imaging) of biological and technological samples.

The proton spectrum in our study is quasi-monoenergetic around the 10 MeV level and contains easily enough protons to produce images in a single shot. Protons are emitted in a reasonable divergence angle around the needle, supporting imaging of multiple objects having a meaningful size (cm-scale). The large proton flux and the large divergence angle are features quite typically observed in laser-driven ion accelerators[50,51], and difficult to achieve with conventional accelerators. In addition, conventional accelerators naturally have very narrow bandwidths and require additional spectral shaping to introduce some energy spread (which will impact other properties like source size)[52]. Meanwhile, most laser ion accelerators feature very broadband

spectra[50,51]. As described earlier in this paper, a monochromatic beam would provide only black-and-white contrast, while a flat-top spectrum would provide nearly no contrast; the presented proton source strikes a nice balance of contrast and dynamic range (in terms of sample thickness). A quasi-monoenergetic spectrum in combination with a detector that is sensitive only in a specific bandwidth (e.g., 10.5–11.5 MeV) facilitates proton-radiographic imaging of a range of sample-thicknesses with a well visible contrast, usable even for complex biological objects, without requiring any further preparations (of source or sample). Note that only very few laser-driven proton sources to date have succeeded in producing beams >10 MeV with similar limited bandwidth and particle counts[36,37], which here directly relates to dynamic range and image contrast. Thus, the proton source itself is competitive in its specification with state-of-the-art laser-driven ion sources.

Finally, using this combined x-ray and proton micro-source, images in both modalities have been recorded within a single laser-shot and have been applied successfully to biological and technological samples. The unique geometrical and spectral properties of the micro-source, including the spatial separation of the imaging beams (protons and x-rays) from the fast electrons and transmitted laser pulse, are the key to recording these bi-modal radiographic images. Future experiments could drive these geometrical and spectral aspects even further, e.g., using fully isolated spherical micro-targets[53] and higher laser intensities.

Based on these experiments, it is worth to discuss perspectives for this kind of source. The x-ray image is automatically registered with the proton image. Interesting use-cases could arise wherever the direct local proton stopping power of a sample is of relevance, while the image resolution of proton images will always suffer from multiple Coulomb scattering. In such cases, the high-resolution x-ray image can be a valuable addition, allowing to capitalize on the complementary strengths of charged particle and photon imaging. Arguably the most promising example in this respect is the combination of both images in post-processing, which has been shown to be useful in the planning steps of charged particle therapy of cancer[7,8]. In this field, to date, the largest error stems from the difficult conversion from x-ray CT to the therapeutic ion irradiation-plan[54]; the presented source can serve as a compact test environment for small animal, preclinical, and clinical studies in these imaging applications. Similarly, the laser-driven micro-source lays groundwork towards laser-driven image guided cancer therapy with a compact machine, using x-rays for imaging and protons for tumor irradiation and/or ima-ging[55]. At present, such ideas are pursued with bigger conventional machines, i.e., refs. [56,57]. In such applications, the automatic spatial and temporal registration of protons and x-rays eliminates a significant portion of uncertainties, e.g., stemming from object deformations or movements between image acquisitions with different modalities, and from related post-processing errors. This could nurture developments in adaptive radiotherapy by significantly shortening the feedback loop, which would also open directions for optimization via machine learning[58]. At last, owing to their brightness and ultrashort temporal characteristics (ns at meter scale), laser-driven ion beams have recently been discussed as promising candidates for FLASH radiotherapy, which leverages the different biological response of cancer and regular cells to such beams[59,60]. In these ultra-high dose rate scenarios, an integrated imaging capability can be of particular interest.

In addition, both sources are automatically synchronized with the optical high-power laser pulse, having well defined timings with respect to one another (depending on the source-sample distance, down to ps-range at 100 μm) and timing jitters smaller than the laser pulse duration. Consequently, another application

for such a source could be pump-probe experiments; e.g., observing density and field distributions in a plasma simultaneously and unambiguously, which has previously been done only with single species sources[19,31,32].

For full fruition, these perspectives require further developments in stability, laser-, target-, and detector-repetition rates, particle-energy and image evaluation in order to become reality. As these needs are in common with many other applications for laser-driven and conventional sources, they are currently being addressed in various laboratories. Petawatt lasers operating at 1 Hz repetition rate are readily available[61], systems at 5–10 Hz are coming online and advanced target designs allow to exploit these repetition rates[62,63]. Higher laser energies providing higher beam energies are not expected to cause any more fundamental problems in the micro-target based approach to simultaneous bi-modal imaging. Fast image detection using time of flight detectors[64], scintillator screens with gated cameras[65], and others are ongoing developments.

## Methods

**Laser, target, and beam generation.** The Texas Petawatt laser (TPW) with wavelength of 1.056 μm, FWHM pulse duration of 150 fs, and energy of 100 J is focused by an off-axis parabolic mirror to a FWHM focal spot size of 5 μm containing ~50% of the nominal laser energy, and used as driver in our experiments. Upstream of the focus, the temporal distribution of the pulse is further cleaned by an in-line plasma mirror with a throughput of 80% (i.e., 80 J arrived at the target of which 40 J were contained within 5 μm) yielding a contrast ratio of ~$10^{-10}$ up to 100 ps before the peak interaction, so as to prevent effects of pre-pulses on the target as far as possible.

We employ commercial tungsten needles with a converging taper at their tip as targets, such as used in scanning tunneling microscopy (Bruker TT-ECM10). For optimum conversion efficiency we focus the laser pulse onto a position at the upright standing needle, which matches the size of the focal spot, i.e., several 10 μm below the tip. The linear laser polarization points in vertical direction and is aligned with the target. We monitor the transmitted laser-beam intensity profile in the near field (supplementary note 3). For shots on needle targets, the original projection-outline of the beam is still visible. Shots, which happen to be well aligned with the micro-target (owing to the shot-to-shot pointing fluctuation), produce beam attenuation in a vertically extended area across the central horizontal region of the laser-beam projection. Regions towards the horizontal edge of the original laser projection are less attenuated, confirming that large parts of the non-depleted laser energy are transmitted in the original laser propagation cone.

Emission of x-rays is expected to occur via free-free, free-bound, and bound–bound interactions in the full solid angle. Although the employed method does not resolve fine spectral structures, free-free emission (Bremsstrahlung) is expected to dominate with the current material choice. As a consistency check it shall be noted that straight-forward calculations assuming other spectral forms (mono-energetic or flat-top) did not reproduce the measured filter transmission. Emission of protons and other ions can be expected to occur in a limited polar angle around the horizontal plane[66] in a full azimuthal angle of 360°, in analogy to the target normal sheath acceleration of ions in planar targets. Sufficient numbers of protons are present on the tungsten-target surface in form of thin (few nm) hydro-carbon contamination layers[67]. Towards the sides, they explode away from the target owing to positive charge surplus created in the target, by electrons that are stripped by the laser's ponderomotive potential. The small spatial extent (layer thickness) can be converted to a small spectral bandwidth towards the sides[38]. In the laser propagation direction, stripped fast electrons seem to cause slightly higher ion energies at cost of broadened ion spectra, as typically observed in laser thin-foil interactions. The maximum kinetic energy of protons for the needle target is reduced compared with foil targets. This is expected owing to the geometrical effect of the needle, which distributes the field-energy in a larger space (i.e., full plane for the needle vs. narrow emission cone for the foil target).

**Detectors.** X-ray spectra, source size, and images were recorded with BAS-TR type imaging plates by Fuji Film (IP) and scanned with a calibrated Typhoon FLA7000 scanner at a nominal 25 μm resolution. The true resolution in the detector plane was found to be slightly worse than that (ca. 50 μm). Taking into account the projection magnification of 17.5–27, the effective spatial resolution for source measurements is 1.85–2.85 μm. FLUKA simulations of the source size measurement were done for x-rays (1–10 keV) and for protons (separate for 4–5 MeV and 35–38 MeV) indicating a similar resolution. If a second IP read-out was required to omit saturation, an almost signal-independent factor of 2.4 has been determined and applied in order to scale images from the second read-out to the original signal-level. Ion spectral distributions were recorded with IP. Proton radiographic images were recorded with CR39 detectors that were etched in six-molar NaOH solution for 20–50 minutes at 80°C. The resulting ion tracks were

recorded using a photographic technique similar to Gautier and Paudel[68,69] or by an automated dark-field microscope by Zeiss at ×10 magnification and evaluated by automated registering and counting of single tracks. Which of both methods was used is specified in the corresponding figure captions. The energy ranges visible in the CR39 detector were calibrated by recording spectrally dispersed traces in the spectrometers on top of the calibrated IP detectors (cross-calibration).

**Particle-in-cell simulations**. Particle-in-cell (PIC) simulations have been performed with the open source code PIConGPU 0.5.0-dev-60ad9eb85[70,71] using its OpenMP backend on the Cori Intel Knights-Landing partition at NERSC. The target has been approximated with 2D3V simulations (10 million NERSC core-hours each) for s- and p-polarization, implying periodicity along the needle axis. As a reference density with feasible computational costs, a tungsten target with 5 micron diameter is assumed to be ten times pre-ionized ($n_e = 634n_c$) and its surface is covered with a pre-ionized 4 nm carbon-hydrogen layer ($n_e = 350n_c$). The spatio-temporal resolution was chosen with quadratic cells of size $dx = 1.16$ nm (910 cells per laser wavelength) and time step $dt = 2.73$ as (at 99.9% of the CFL criteria; resolving the peak electron frequency with $\omega_{p,e} \cdot dt = 0.12$). The Yee-solver has been used as field solver, particles are modeled with TSC (2nd order) B-splines, 20 particles per species, and cell (40 particles per cell for the tungsten needle; 80 particles per cell for the contamination layer), Esirkepov current deposition, Boris particle pusher, and trilinear force assignment. The laser pulse is modeled as Gaussian beam with 1.06 μm central wavelength, 150 fs pulse length (FWHM of intensity), 5 micron focal spot size (FWHM of intensity) focused to the target center with a normalized peak amplitude of $a_0 = 45$. In situ reduction methods (PIConGPU plugins) have been used for presented diagnostics, filtering particles within ±2 degree pointing along respective observation axes. Time is given relative to peak intensity on target and Fig. 1e presents p-polarized data (further data in supplementary note 4). Proton energy spectra are evaluated before particles leave the computational box and compressed target density is plotted two pulse lengths after peak intensity on target. S-polarized data are available in supplementary note 4.

## Data availability

The data that support the findings of this study are available from the corresponding authors upon reasonable request. PIC simulation input files, plot scripts, in situ reduced diagnostics data, and exact PIConGPU source code of the simulation are archived at: https://doi.org/10.5281/zenodo.3630701. Full-resolution particle and field data are archived at the NERSC computing center on HPSS tape drives and are available upon reasonable request.

## Code availability

PIConGPU is developed as open source code with its complete change set history and documentation available at https://doi.org/10.5281/zenodo.591746.

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

## Acknowledgements

This work was supported by the DFG via the Cluster of Excellence Munich-Centre for Advanced Photonics (MAP) and Transregio SFB TR18. T.M.O. acknowledges support from IMPRS-APS. This work has been carried out within the framework of the EUROfusion Consortium and has received funding, through the ToIFE, from the European Union's Horizon 2020 research and innovation program under grant agreement number 633053. A.H. acknowledges support by the Exascale Computing Project (17-SC-20-SC), a collaborative effort of two U.S. Department of Energy organizations (Office of Science and the National Nuclear Security Administration). T.M.O. and A.H. acknowledge resources of the Lawrence Berkeley National Laboratory and the National Energy Research Scientific Computing Center (NERSC), which are supported by the U.S. Department of Energy Office of Science under contract no. DE-AC02-05CH11231. The authors acknowledge funding by the Air Force Office of Scientific Research (AFOSR) (FA9550-14-1-0045, FA9550-17-1-0264). The authors thank the TPW facility staff for operating the laser and their strong support during the whole campaign. The authors thank Cameron Geddes, Paul Neumayer, Chiara Gianoli, and Matthias Würl for discussions and input.

## Author contributions

T.M.O. and J.S. conceived the experiment idea. T.M.O., C.K., F.S.E., J.H., J.Ge. built the setup and carried out the experiments. G.D., E.M., and G.T. provided support with the local infrastructure at the TPW facility. The TPW laser was built, maintained and operated by E.G., M.E.D., G.D., J.Go., M.S., B.M.H., M.M. A.H. contributed particle-in-cell simulations. F.S.E. contributed FLUKA simulations of the source size measurement. T.M.O. evaluated and interpreted experimental data with input from J.S., B.M.H., J.W., D.H., P.H., F.S.E., and K.P. T.M.O. prepared the manuscript with additional input from J.S., A.H., D.H., F.S.E., P.H., J.,W., and K.P.

## Funding

## Competing interests

The authors declare that they have no competing interests.
