## [Peer Review File · Nature Communications]

Reviewers' comments:

Reviewer #1 (Remarks to the Author):

This paper presents an experimental setup for realising multi-modal single pulse acquisition radiography – namely proton and x-ray radiography – utilising the emission from a laser-irradiated micro needle tungsten target. Source size and spectral measurements of the emission are presented alongside radiographs of biological and mechanical objects.

While not the first time that multi-modal proton-x-ray radiography has been demonstrated, this is the first application of the technique with samples relevant to communities outside of the laser-plasma community and is therefore an important step to engage with those that can derive impact from this technological development.

There are quite a few points that I believe the authors need to address before recommending for publication in Nature Communications. I suggest that the authors conduct particle-in-cell and radiation transport simulations to explore the proton acceleration and x-ray emission, respectively, in order to validate the physics at play - this will greatly strengthen the source characterisation section and the quality of the paper.

In future, please consider numbering the lines of the manuscript, to make it easier for reviewers to make reference to specific parts.

Abstract:

Need to make a snappier case for simultaneous imaging – specifically to proton and x-ray imaging. Why are these two modalities, in particular, complimentary? Why is there need for short pulse single shot acquisition? Why is this work of such importance and what impact can it have? You have the content for this scattered about in the manuscript but I think it will make for a much stronger and attention-grabbing abstract to see these points summarised here.

Page 1:

“...construction of the first laser, laser-driven particle accelerators and secondary light sources...”

Were these very early observations really due to accelerator dynamics, or were they thermal explosion dynamics? I think it's safer to just delete the word 'accelerators' here as the sentence has the same impact without it and 1966 interaction physics is quite distinct from the laser-accelerator physics of topic in the present work.

“Laser-plasma sources have been considered and explored to replace conventional single species....”

I strongly recommend that you reconsider your wording here, perhaps go for “...and explored as complimentary to conventional....” or “...have been considered and explored for many applications, mostly for their....” as the current phrasing is a sure fire way to disengage anyone from those single-

species communities, especially at a time when we need to be engaging with these application communities for development and transfer of this technology. Perhaps when we have commercial ready prototypes is when we can start describing these as replacement technology.

“Respective other radiation modalities.....”

I think this is a negative statement that isn't productive for the paper, and instead distracting, and the whole sentence can be omitted. Instead, consider making a punchy sentence to introduce the positive impact of your work. “Here we demonstrate the benefit of multi-modal emission from intense laser-matter interactions and propose a target designed to optimise the emission characteristics for advanced biological imaging.”

“In addition, with more laser energy being focused to the same spot as a lower energy laser, the target-foil-area which is driven by”

You need to justify this statement by including a sentence discussing the effect of increasing laser intensity (by increasing laser energy) on the divergence of the laser-accelerated fast electrons, with reference to previous work (e.g. J. Green PRL 2007 paper plotting divergence increasing with laser intensity).

“...., featuring a 20%-level energy spread (FWHM).”

How are you getting this spectral feature? I know you go on to explain it later in the Methods section but I am so distracted by asking “why?” to this statement as it is not yet a normal feature for laser-driven ion beams. All it needs is a short addition to the sentence saying that it's due to Coulomb explosion dynamics of the contaminant layer being the dominant acceleration mechanism for side emission.

Page 2:

Nice clear experiment layout diagram!

Page 3:

“....benefits the purpose of radiographic imaging in the proposed scheme.”

Why? Why is limited bandwidth protons useful in proton imaging? Please expand on this here when you first introduce it.

“...the source can maintain a small size despite the higher laser energy”.

Need to justify this statement with the physics behind it. Consider adding “due to fast electron transport, and therefore x-ray generation, being confined to the volume of the narrow wire target.”

“...a fan-bean...”

Typo :)

“... = 2360 ± 140 eV...”

This is a really low x-ray temperature for the laser interaction conditions. What's the x-ray generation mechanism at play? Is the fast electron temperature at these wide angles to the laser propagation really that low? Even k-alpha temperature for tungsten is much higher than this. This really needs addressing here or in the discussion and is why you need to carry out a simulation study to really explore what is going on here with these needle targets. It is very apparent that the physics explanations are missing in this paper.

“...peaked energy spectrum beyond 10 MeV has long been desired from a laser driven accelerator.”

Why? For broad readership of Nature Communications it is important to be very explicit as not everyone will know.

“...by sharp silicon edges of few mm effective...”

How did you come to the decision to use this material and this thickness? Did you do the x-ray transmission calculations to determine what fraction of x-rays are attenuated by this target to ensure it gives a 'black' edge. These little details really matter in a paper like this.

“With extent in the 2-5 μm -range, ...”

How is the x-ray and proton source in the horizontal smaller than the width of the needle? What is going on here? It really is mind-blowing that the x-ray generation is confined to a region smaller than the spot size of the laser and the width of the needle. Is there some sort of focusing or confinement at play? What is the emission mechanism? And if the horizontal emission is looking at the direction along the laser propagation axis, why are we not seeing x-ray emission along the full width? Why is the generation so localised so to give smaller than 5 micron spot size? So many questions left unanswered. This is to me the most interesting physics part of the paper and if you can nail the explanation with simulations or analytical modelling then it will be a very high impact finding.

Page 4:

“This projection is used to record...”

What is the effective magnification for the x-ray imaging shown in figure 2?

“Increasing magnification for the x-ray image in the point-projection...”

This phase-enhanced data is really stunning and I think you should promote to the main manuscript instead of supplementary material, as it really hammers down the impact of working with these

needle targets and is a great complimentary imaging mode on top of the multi-species imaging. However, you really need to explain the origin of these detector limited micron-scale source sizes to strengthen the portfolio of work shown here and validate your observations.

“...and produced radiographic contrast for...”

Selling yourself short here in this summary statement. Consider adding: “and have been applied for the first time to biological and technological samples.” to highlight the novelty of the work presented.

“..facilitate high-resolution radiographies.”

Can you achieve the ~ 5 micron source-limited imaging resolution AND multi-modal imaging simultaneously? What other factors does one need to consider for exploitation of this technique? How far back does the IP need to be so that you are no longer detector resolution limited and are taking advantage of tiny source size for high resolution? And then is there still enough photon density for single shot acquisition? Is point-projection proton radiography possible? So that you can have the sample close to the source and detectors far enough away to achieve high resolution. And then if the sample is close to the source point is it destroyed by intense radiation exposure? My point being: careful not to oversell here and make sure you manage expectations of laser-driven imaging technology. Remember that it won't be just laser-plasma people reading this paper, so you need to be explicit with the potential but also honest with limitations of the source.

Page 5:

“...could be extended to include neutron radiography in a single shot.”

How are you proposing the neutron emission is generated? If via a mechanism that utilises the ion emission then this negates the possibility for simultaneous proton radiography? I can't see how proton-xray-neutron radiography is a compatible extension of this work. Unless the neutrons are from in-target reactions and the protons are from surface coulomb explosion as in present work? But I would love to be convinced otherwise if you can provide a brief extended discussion to this statement for the manuscript. Neutron-xray radiography on the other hand is absolutely compatible (if you can get your 2 keV x-rays through the neutron convertor material or if you shoot a deuterated needle target perhaps?) and is a very complimentary imaging modality.

“...small footprint...”

Typo :)

“With such ideas, advantages of laser-driven sources could be enriched beyond their...”

This is a great statement summarising the unique strengths of laser-driven sources! Very strong justification for the current work. You should promote this sentence to much earlier in the manuscript and put it the abstract also. Maybe even second or third sentence of the abstract.

Page 6:

"....repetition rate are readily available."

And 5 - 10 Hz systems are currently coming on line too.

"The linear laser polarization..."

So it's parallel with the needle surface and therefore S polarisation? Or does the needle have a tapered edge/fast gradient taper so that the polarisation isn't exactly parallel? Or in the area of the laser spot can we assume the needle acts like a vertical wire and therefore laser E-field is parallel to surface?

"Towards the sides, they explode..."

So in the region of interest and where you saw peaked spectrum the ion acceleration at play here is Coulomb explosion acceleration? Can you perform simulations to confirm your discussion here and validate your interpretation of the physics of this laser-needle interaction? This really feels like it is missing from the paper. Also, I think that this part should be promoted to the discussion section of the main manuscript and not hidden in methods as it explains the key physics of these interesting, and necessary for applications, spectra.

"...magnification of 17.5-27....."

I take it this was the magnification just for the phase contrast/high resolution shots? The multi-modal shots were with far lower magnification as the IP was only 0.5 m from the CR39. How far back was the IP for this high magnification projection imaging? And you still managed to get phase contrast imaging in a single shot? What is the divergence angle of the x-ray emission? Did it show any collimation?

Reviewer #2 (Remarks to the Author):

The major claim in the paper is a simultaneous single-shot radiographic imaging technique with a laser-driven x-ray and proton micro-source.

The proposal to use both x-rays and protons in a simultaneous radiographic imaging technique is not novel. The idea was reported by Nishiuchi et al., Journal of Physics: Conference Series 112, 042036

(2008). The novelty in the reported work concerns the micro-needle target that is used, which solves the outstanding problem of a simultaneous small virtual source for protons and a small x-ray spot size.

The paper will be of interest to others in the field and shows the value of this target interaction for bimodal radiographic applications. The paper will influence thinking in the field through the development and optimization of novel target interactions to engineer multi-modal and ultrafast radiographic techniques. The claims are convincing and the characterization techniques used appear sound.

The result would be more impactful with a dynamic demonstration to highlight the ultrafast nature of the bimodal radiographic technique. This aspect is not tested in the reported work, but represents an important part of the significant promise of ultrafast, laser-based sources (bimodal or otherwise). Such tests may be challenging, but feasible by splitting the short pulse beam to generate a driver to excite a dynamic radiographic object.

The claims are appropriately discussed in the context of previous literature. The manuscript is clearly written and they have not oversold their claims. They have been fair in the treatment of previous work and sufficient detail has been provided that the work could be reproduced.

Reviewer #3 (Remarks to the Author):

The paper "Simultaneous single-shot radiographic imaging with a laser-driven x-ray and proton micro-source" by Ostermayr et al., describes the use of a single intense laser pulse to generate sources of protons and x-rays for radiography. The experimenters use a low repetition rate high energy laser incident onto a tungsten microneedle target. This produces an interesting proton source, emitted at all angles as well as an x-ray source also emitted at all angles due to bremsstrahlung or possibly atomic emission. There is no theoretical discussion and no modelling of the interaction.

The paper involves the use of these radiation sources for radiography of a biological object simultaneously with both x-rays and protons.

Positive aspects of this work are:

- 1) The authors measure an unusual peaked proton spectrum in the transverse direction, although this is unexplained.
- 2) An unusually small source size of x-rays in transverse direction is measured. There is both bremsstrahlung and atomic emission from such interactions which would have different directional, source size and pulse duration characteristics. There was not much data on source size variation with laser properties and target properties presented in the paper.

Problematic aspects of this work are:

- 1) X-ray phase contrast imaging and proton imaging have been done previously using solid targets

although separately. The justification for why one would want to do both at the same time in this geometry is not convincing.

2) It would obviously provide better and more controllable simultaneous radiography data by just splitting the Petawatt beam in two and having two optimized sources with temporal control. This could also be done with any existing multibeam short pulse system (ARC, OMEGA, ORION etc.)

3) The main scientific interest in proton radiography is in measuring time resolved fields, however here the protons are used to measure only the density providing the same information as the x-rays but with less resolution. For field measurements with a single beam as the source the x-ray and proton measurements would be separated in time by many picoseconds so this wouldn't be necessarily be useful.

4) Why is the scattered light only in the forward direction in the experiment? This suggests that the microneedle target is destroyed by the pre-pulse before the interaction and the interaction is with an underdense plasma.

5) What is the spectrum of the x-rays? How are the components due to atomic emission and bremsstrahlung different with respect to source size, spectrum etc.

In conclusion this is an interesting paper however I don't believe that it is of sufficient importance to be published in Nature Communications. It is actually more appropriate for Review of Scientific Instruments or Applied Physics Letters. For a high impact journal such as Nature Communications the authors would need to use this source to make a new scientific measurement which couldn't have been done in another way. They also need to do a more in-depth investigation of the micro-needle interaction which would include numerical modeling.

Dear Referees,

We would like to thank you for your efforts in reviewing this manuscript. In the following, we respond to comments and provide snapshots of changes to the manuscript, with removed text highlighted in small red font, and added text highlighted in blue sans serif font, respectively.

Reviewer #1

Overview comment (1): This paper presents an experimental setup for realising multi-modal single pulse acquisition radiography – namely proton and x-ray radiography – utilising the emission from a laser-irradiated micro needle tungsten target. Source size and spectral measurements of the emission are presented alongside radiographs of biological and mechanical objects.

While not the first time that multi-modal proton-x-ray radiography has been demonstrated, this is the first application of the technique with samples relevant to communities outside of the laser-plasma community and is therefore an important step to engage with those that can derive impact from this technological development.

There are quite a few points that I believe the authors need to address before recommending for publication in Nature Communications. I suggest that the authors conduct particle-in-cell and radiation transport simulations to explore the proton acceleration and x-ray emission, respectively, in order to validate the physics at play - this will greatly strengthen the source characterisation section and the quality of the paper.

Our response: We thank the referee for this careful review and positive appreciation of our work. We appreciate the great effort in helping us to improve the manuscript. Below, you find our replies to your comments and related changes to the manuscript including new simulations and data. In addition, we took a critical look at our experimental data and re-evaluated large parts after identifying a small mistake in the linearization of IP data, which however, left our essential findings and the story unchanged. We hope you find our revised and substantiated manuscript appropriate for publication in Nature Communications.

Referee comment 2: In future, please consider numbering the lines of the manuscript, to make it easier for reviewers to make reference to specific parts.

Our response: Duly noted. Added line numbering.

Referee comment 3: Abstract:

Need to make a snappier case for simultaneous imaging – specifically to proton and x-ray imaging. Why are these two modalities, in particular, complimentary? Why is there need for short pulse single shot acquisition? Why is this work of such importance and what impact can it have? You have the content for this scattered about in the manuscript but I think it will make for a much stronger and attention-grabbing abstract to see these points summarised here.

Our response: We thank the referee for this advice and his observation of these points within the paper. We extended our abstract, trying to collect and highlight these points.

Changes to the manuscript:

Novel imaging methods have revealed some of nature's most fascinating and unexpected secrets. Today, radiographic and Radiographic and tomographic imaging with x-rays and protons are an omnipresent tool in basic research and applications relevant to industry, material science, military and medical diagnostics. In some of these examples cases, the information contained in both modalities is can be valuable in principle, but difficult to access simultaneously. Laser-driven solid-density plasma-sources have long been known to deliver both kinds of radiation at the same time, but mostly single modalities have been explored for applications, aiming to replicate use-cases of conventional single-species sources. Their potential for bimodal bi-modal radiographic imaging has never been fully realized, due to conceptual problems in generating appropriate sources and recording well separated images. Here, we report on the generation of proton and x-ray sources micro-sources in a laser-plasma interaction of the focused Texas Petawatt laser with a solid-density, micrometer-sized Tungsten tungsten needle. We demonstrate their unique capabilities in terms of spectral and spatial distributions and use them for bimodal apply them for bi-modal radiographic imaging of biological and technological objects in a single laser shot. The small x-ray source additionally enables images with phase-contrast contribution. Bi-modal imaging could serve as a compact test environment for studies of the conversion process from x-ray attenuation images to proton stopping maps. The combination of both images via post-processing could use the higher spatial resolution of the x-ray image, which does not suffer from multiple Coulomb scattering, to improve the proton image. Such approaches can directly benefit from the single-shot bi-modal imaging, where both images are automatically registered on top of each other in time and space. The short pulse lengths and the fixed temporal relation between protons and x-rays could benefit recordings of moving objects, e.g., living/breathing organisms or plasma instabilities. With such ideas, advantages of laser-driven sources could be enriched beyond their small footprint by embracing their additional unique properties, including the spectral bandwidth, small source size and multi-mode emission.

Referee comment 4: "...construction of the first laser, laser-driven particle accelerators and secondary light sources...."

Were these very early observations really due to accelerator dynamics, or were they thermal explosion dynamics? I think it's safer to just delete the word 'accelerators' here as the sentence has the same impact without it and 1966 interaction physics is quite distinct from the laser-accelerator physics of topic in the present work.

Our response: We agree and changed the manuscript accordingly.

Changes to the manuscript:

28 ing combined images. In parallel to advances in imaging,
29 shortly soon after the construction of the first laser⁹, laser-
30 driven particle accelerators¹⁰ and secondary light sources¹¹
31 have emerged been studied.

Referee comment 5: "Laser-plasma sources have been considered and explored to replace conventional single species....."

I strongly recommend that you reconsider your wording here, perhaps go for "...and explored as complimentary to conventional....." or "...have been considered and explored for many applications, mostly for their...." as the current phrasing is a sure fire way to disengage anyone from those single-species communities, especially at a time when we need to be engaging with these application communities for development and transfer of this technology. Perhaps when we have commercial ready prototypes is when we can start describing these as replacement technology.

Our response: We agree that our initial phrasing could have been perceived offensive by some readers, which was not our intention. We opted for the second suggestion.

Changes to the manuscript:

38 Laser-plasma sources have been considered and ex-
39 plored to replace conventional single species sources for some for
40 many classical single-source applications, mostly for their
41 promise to provide a more compact source based on
42 higher applicable field strengths¹⁶⁻²¹. Respective other ra-

Referee comment 6: “Respective other radiation modalities....”

I think this is a negative statement that isn't productive for the paper, and instead distracting, and the whole sentence can be omitted. Instead, consider making a punchy sentence to introduce the positive impact of your work. “Here we demonstrate the benefit of multi-modal emission from intense laser-matter interactions and propose a target designed to optimise the emission characteristics for advanced biological imaging.”

Our response: We agree and largely followed the referee's advice here.

Changes to the manuscript:

42 higher applicable field strengths¹⁶⁻²¹. Respective other ra-
43 diation modalities, which naturally co-exist in certain laser-plasma in-
44 teractions, have often been regarded as a disturbance that must be
45 suppressed or eliminated in the radiation-generation and within the
46 application. Here we demonstrate the benefit of multi-modal
47 emission from intense laser-matter interactions and propose
48 a target designed to optimize the emission characteristics
49 for advanced imaging.

Referee comment 7: “In addition, with more laser energy being focused to the same spot as a lower energy laser, the target-foil-area which is driven by”

You need to justify this statement by including a sentence discussing the effect of increasing laser intensity (by increasing laser energy) on the divergence of the laser-accelerated fast electrons, with reference to previous work (e.g. J. Green PRL 2007 paper plotting divergence increasing with laser intensity).

Our response: We included reference (J. Green, PRL 2008) as an additional aspect for understanding the larger hot electron area (and hence x-ray source) with higher peak intensity.

However, we think the most apparent and straight forward explanation for increased source size (the target area illuminated by sufficient intensity for x-ray emission increases with peak intensity) as given in the text remains another valid point. Even without divergence considerations.

Changes to the manuscript:

72 ous. In addition, with more laser energy being focused to
73 the same spot size as a lower energy laser, the target-foil-
74 area which is driven by sufficient intensity to emit protons
75 and x-rays would naturally increase. This would cause is naturally
76 increased. Similarly, the divergence of hot electrons gen-
77 erated in the laser-plasma interaction increases with laser
78 intensity²⁶. These effects would result in a larger source
79 size and reduced image resolution. While a small vir-
80 tual source for protons can often be maintained due to
81 their laminar flow²⁷, the x-ray spot size depends on this
82 effect²⁸.

Referee comment 8: “...., featuring a 20%-level energy spread (FWHM).”

How are you getting this spectral feature? I know you go on to explain it later in the Methods section but I am so distracted by asking “why?” to this statement as it is not yet a normal feature for laser-driven ion beams. All it needs is a short addition to the sentence saying that it's due to Coulomb explosion dynamics of the contaminant layer being the dominant acceleration mechanism for side emission.

Our response: Done. Added the explanation from later in the manuscript right here.

Changes to the manuscript:

83 Here, we present and characterize a novel laser-driven
84 source, based on a micro-needle-target, that solves these
85 issues in a combined approach. The ~~specral~~ **spectral** char-
86 acteristics of protons emitted towards the side is found to
87 be peaked around 10 MeV, featuring a 20%-level energy
88 spread (FWHM). ~~This benefits the~~ **The protons stem from**
89 **a nm-thin CH contaminant layer present on the Tungsten**
90 **surface; the limited spatial extent translates to a limited**
91 **spectral bandwidth observed towards the side, as the pro-**
92 **tons explode away from the positively charged target into**
93 **vacuum.**

Referee comment 9: Nice clear experiment layout diagram!

Our response: Thank you!

Referee comment 10: "...benefits the purpose of radiographic imaging in the proposed scheme."

Why? Why is limited bandwidth protons useful in proton imaging? Please expand on this here when you first introduce it.

Our response: We moved the explanation and example calculation from later in the manuscript up to this point.

Changes to the manuscript:

94 Spectra with a limited bandwidth (as opposed to mo-
95 noenergetic or ultrabroad spectra) can benefit the purpose
96 of proton radiographic imaging ~~in the proposed scheme.~~ **with**
97 **a suitable detector that is sensitive in a specific spectral**
98 **range (in a range higher than the spectral peak). Then, a**
99 **measured proton count corresponds uniquely to a specific**
100 **object thickness, because only a specific part of the proton**
101 **spectrum is transmitted through the object in such a way,**
102 **that its residual energy corresponds to the detector's sen-**
103 **sitive bandwidth. In this configuration, the proton's initial**
104 **spectral modulation depth translates directly into achievable**
105 **contrast (neglecting other factors like detector noise), while**
106 **the spectral width determines the range of object thick-**
107 **nesses that can be imaged. These considerations have been**
108 **a challenge for early ion-radiographic imaging¹ based on**
109 **conventional accelerators delivering highly monoenergetic**
110 **(% level) beams. These could produce almost binary radio-**

111 graphic contrast, but only under perfectly matched condi-
112 tions for the imaged object. On the other hand, a perfectly
113 flat spectrum would not produce radiographic contrast at
114 all, because each sample thickness would be represented by
115 the same particle count in the detector. Contrasting these
116 two extreme scenarios, spectral distributions as measured
117 here feature some, but not an infinite, spectral bandwidth.
118 Hence they allow proton radiographic imaging in a range
119 of object thicknesses with reasonable contrast without ad-
120 ditional preparations of the object, beam or detector. For
121 example, a proton spectrum that decays linearly from its
122 maximum at 10 MeV towards zero count at 15 MeV, and
123 a detector that is sensitive between 10 and 11 MeV, will
124 produce contrast for objects in a water equivalent thickness
125 range from 0 to about 1 mm.

Referee comment 11: "...the source can maintain a small size despite the higher laser energy".

Need to justify this statement with the physics behind it. Consider adding "due to fast electron transport, and therefore x-ray generation, being confined to the volume of the narrow wire target."

Our response: We added an explanation, which besides points mentioned by the referee includes effects caused by the curved target surface and the plasma expansion into vacuum. We will come back to this in response to comment 16.

Changes to the manuscript:

139 the image. ~~Due to the few- μm dimensions of the target, the~~ **The**
140 **source can maintain a small size despite the higher laser**
141 **energy. , due to the confinement of fast electron transport,**
142 **and therefore x-ray generation, to the small volume of the**
143 **needle target. Two effects can further reduce this volume:**
144 **the curved target surface can lead to a converging plasma**
145 **movement in the laser interaction, and the plasma expansion**
146 **into vacuum can reduce the high density region effectively**
147 **by losing part of the plasma to lower density regions.**

Referee comment 12: "...a fan-bean..."

Typo :)

Our response: Corrected.

Referee comment 13: "... = 2360 ± 140 eV..."

This is a really low x-ray temperature for the laser interaction conditions. What's the x-ray generation mechanism at play? Is the fast electron temperature at these wide angles to the laser propagation really that low? Even k-alpha temperature for tungsten is much higher than this. This really needs addressing here or in the discussion and is why you need to carry out a simulation study to really explore what is going on here with these needle targets. It is very apparent that the physics explanations are missing in this paper.

Our response:

First, please note that the temperature after re-evaluation was found as 8060 ± 940 eV. This was the major change introduced by our IP re-evaluation. The (still) low temperature may be astonishing at first. But we think that, instead, it can be explained as a case of 'you will only observe, what your detector can measure'.

In fact, only few percent of the electrons, primarily from the target surface, will be promoted to become hot (ponderomotively driven) electrons. The existence of much lower energy populations in the bulk (i.e. larger quantities) has been established, e.g., by Kemp and Divol.

<https://aip.scitation.org/doi/10.1063/1.4963334>

In our specific setup, we used filters of up to 430 um thick aluminum to determine the x-ray spectrum, with imaging plates having a peak sensitivity around 6 keV. Images were recorded in a matching setup in terms of equivalent thickness. This will dictate the range of our spectral sensitivity.

We would like to clarify this with an example: consider we record an x-ray source with an equivalent conversion efficiency of $1e-4$ from laser energy to x-ray energy, for two spectral regions. The first region is 1-10 keV (conversion efficiency as indicated by our experiment) and the second one is 20-200 keV (conversion efficiency as indicated by Borm et al. for comparable laser parameters).

B. Borm et al. Properties of laser-driven hard x-ray sources over a wide range of laser intensities. Physics of Plasmas, 26(2):023109, 2019.

With our setup, the measured signal would always be dominated by the lower spectral range because

a: the same amount of emitted x-ray energy (same conversion efficiency) corresponds to a smaller number of photons in the higher energy band

b: the spectral response of the IP (quantum efficiency, or signal counts per photon) is lower in the higher energy range

In other words, the higher energy band may be present in the current setup's emission, but not contribute significantly to the measured signal.

Then, the real question is what mechanism generates the x-rays in this low (1-10 keV) spectral range? More information regarding their generation can be obtained in experiments with direct spectral resolution (in contrast to retrieval algorithms based on a specific distribution as used in our paper). Our collaborator Dr. Paul Neumayer (GSI Darmstadt) kindly shared data from shots with the PHELIX laser (up to 180 J, 5 um FWHM focus diameter, 500 fs FWHM duration, 5 um focus) onto 5 um thick tungsten wires. This is a fairly comparable scenario to ours. X-ray spectra were recorded with a HOPG spectrometer.

Figure 1: (top) Low energy (mJ) shot on tungsten wire at PHELIX laser. (bottom) High energy (180 J) shot on tungsten wire at PHELIX showing Bremsstrahlung continuum plus broad ‘lines’. Note: all scales are linear. Figure by courtesy of Dr. Paul Neumayer.

A low energy shot (1 mJ) shows almost exclusively the tungsten L alpha and L beta lines. At full energy of 180 J, the spectrum consists of a continuum to which a temperature of ca. 3.5 keV can be fitted, and some smeared out structures on top of that, which have been identified as transitions into the L-shell of highly ionized tungsten. This explains why our simple Bremsstrahlung fit performs reasonably well. A short version of the above discussion is also part of

- [1] L. Antonelli et al., X-ray phase-contrast imaging for laser-induced shock waves. EPL (Europhysics Letters), 125(3):35002, Feb 2019.

Which we added as citation to the manuscript.

Changes to the manuscript:

183 tion factor. With a fit temperature of $k_B T_e = 2360 \pm 140$
 184 $k_B T_e = 8060 \pm 940$ eV (95% confidence level) and con-
 185 sidering emission in the full solid angle, this spectrum
 186 contains an energy of 8.9 1.6 mJ, of which more than 3.8
 187 1.2 mJ are emitted at energies higher than 2 keV. It shall
 188 be mentioned that recent experiments at comparable condi-
 189 tions using tungsten wire targets^{33,34} found similar spectra,
 190 with high-resolution HOPG spectrometers additionally iden-
 191 tifying broadened emission structures around 8.2-8.4 keV
 192 from electron transitions into the L-shell of highly ionized
 193 tungsten. However, the Bremsstrahlung content was iden-
 194 tified as dominant, consistent with our fit. Note that our
 195 detector setup is laid out for these relatively low-energy x-
 196 rays. Higher energy MeV-scale x-rays may also be emitted
 197 from the laser-plasma interaction, but are not expected to
 198 significantly contribute to the recorded signal.

Referee comment 14: “...peaked energy spectrum beyond 10 MeV has long been desired from a laser driven accelerator.”

Why? For broad readership of Nature Communications it is important to be very explicit as not everyone will know.

Our response: We agree with the referee and changed the manuscript accordingly.

Changes to the manuscript:

238 scale. A proton source with a peaked energy spectrum
 239 beyond 10 MeV has long been desired from a laser driven
 240 accelerator with significant progress only compact laser driven ac-
 241 celerators, because it would boost their relevance for many
 242 applications originally developed with conventional acceler-
 243 ators, which per default emit a very narrow proton energy
 244 spectrum. Possibly the most impactful application for a
 245 tightly peaked proton spectrum, is the radiotherapy of cancer,
 246 where the sharp dose deposition at the Bragg peak
 247 is used for targeted dose delivery to tumors. Significant
 248 progress towards peaked spectra beyond 10 MeV have only
 249 been reported in recent efforts exploring advanced acceler-
 250 ation mechanisms³⁵. In the present setup, the formation of
 251 a peaked spectrum is facilitated by a combination of space-
 252 charge effects between different ion species (highly ionized
 253 tungsten ions and protons) and the localization of protons
 254 in a very thin (nm-scale) contamination layer on a needle-
 255 target that does not exceed the focus in size; both effects are
 256 known to facilitate quasi-monoenergetic ion spectra^{36–38}.

Referee comment 15: "...by sharp silicon edges of few mm effective..."

How did you come to the decision to use this material and this thickness? Did you do the x-ray transmission calculations to determine what fraction of x-rays are attenuated by this target to ensure it gives a 'black' edge. These little details really matter in a paper like this.

Our response: The motivation to use silicon was its availability in large quantities, as wafers, at affordable prices with specified atomically flat surface.

In experiments the needle was first aligned with the silicon surface plane using a laser diode. It was then displaced by 50-100 μm normal to the silicon-plane in order to ensure that the second (distal) edge of the silicon would create the shadow for the edge-spread function measurement. Thereby, this small apparent 'misalignment' allowed for the unambiguous definition of magnification ($M=D/L$) by exactly defining the edge position. A schematic of the setup is shown in Figure 2a.

Figure 2: FLUKA simulations of source size measurement. a: setup sketch and measurement idea via edge spread function. b: Simulation for 1-10 keV x-rays from a uniform 1 μm source with 50 μm offset between needle and silicon surface plane. The points are simulated data, red line is the erf-fit, gray area indicates the retrieved FWHM size. c: same setup with 4-5 MeV protons. d: same setup with 35-38 MeV protons.

FLUKA simulations of this setup were performed for x-rays (1-10 keV range) and protons (4-5 MeV and 35-38 MeV ranges) for a lateral offset of 50 μm between needle and silicon surface (Figure 2b-d). The

simulated source distribution had a 1 μm diameter with uniform/isotropic emission. All distances were chosen comparable to our experiments and a 25 μm detector resolution was used. Instead of simulating an iWASP, we simulated two separate energy bands (4-5 MeV and 35-38 MeV) for protons.

Source-size analysis was done in analogy to experimental data, using an erf-fit. In all cases the original 1 μm source is overestimated by the retrieved FWHM size but comes out still well below 2 μm . Parts of the overestimation can be attributed to the simulated uniform source distribution, while the erf-fit assumes an underlying Gaussian LSF/PSF. Another important factor is the limited (but realistic) detector resolution; with about 20-times magnification and 25 μm detector pixel size, the effective detector resolution corresponds to only 1.25 μm (source size)/px.

Meanwhile, the data presented in Fig. 2b-d do not show indications of a 'non-black' edge. Signal levels in the shadow are small for all cases, and the shadow does extend up to the signal rise that is due to the geometrical edge. In other words, in none of the cases significant signal was observed in the geometric shadow (which would complicate measurements).

E.g., traces for 4-5 and 35-38 MeV look comparable, which indicates that higher energy protons still cannot penetrate the edge in significant numbers.

Given these simulations, the method was expected to sufficiently resolve sources in the 1+ μm range. Note that, just as the referee, we did not expect to need such small sources.

Changes to the manuscript: We added the more detailed description of measurements and simulations to the supplementary file.

Referee comment 16: "With extent in the 2-5 μm -range, ..."

How is the x-ray and proton source in the horizontal smaller than the width of the needle? What is going on here? It really is mind-blowing that the x-ray generation is confined to a region smaller than the spot size of the laser and the width of the needle. Is there some sort of focusing or confinement at play? What is the emission mechanism? And if the horizontal emission is looking at the direction along the laser propagation axis, why are we not seeing x-ray emission along the full width? Why is the generation so localised so to give smaller than 5 micron spot size? So many questions left unanswered. This is to me the most interesting physics part of the paper and if you can nail the explanation with simulations or analytical modelling then it will be a very high impact finding.

Our response:

Regarding the measured source sizes that were found smaller than the needle, we note that for protons, no such effect was observed after data re-evaluation.

For the x-rays, we suggest an intuitive idea as to why their source size is small. Targets with curved surfaces are known to facilitate converging movements that can compress the target. The temporally integrated source size will be dominated by the time of highest temperature in the largest number of bulk electrons.

For the converging movement of the bulk, we can estimate its relevance via the hole boring velocity $v_b = 2 a_0 c (Z/A * m_e/m_p * n_c/n_e)^{0.5}$, where Z is the degree of ionization, A ion mass number, m_e is the electron mass, m_p is the proton mass, n_c is the critical density and n_e is the electron density. Integration of v_b from before the pulse to the peak intensity yields a movement for of the initial target surface of

- 250 nm with parameters: $Z=10$, $n_e/n_c=300$
- 900 nm with parameters: $Z=60$, $n_e/n_c=100$.

Another contributing factor is the plasma expansion into vacuum. This expansion reduces the density at the initial target-vacuum interface, and effectively leaves a reduced size of the high-density plasma region, where x-ray generation via Bremsstrahlung is expected to be strongest.

2D3V particle-in-cell simulations (PIConGPU, new fig. 1e main figure) shows agreement with these ideas. We observed both the effect of hole boring and plasma expansion, leading to an overdense region of just 3 μm diameter with an even smaller density enhanced region caused by laser hole boring into the target. Around this high density plateau, density drops exponentially.

In future work, simulations will extend to three dimensions and include further effects, e.g. ionization dynamics, to allow for parameter optimizations. Due to the high ionization levels of Tungsten and consequent electron densities at given laser parameters, 3D simulations require ultrahigh spatial and temporal resolution while also covering enough space and time to capture relevant processes – this effectively limited us to 2D3V simulations in the current work.

Future experimental work at the new Center for Advanced Laser Applications in Garching will aim to better understand, optimize and utilize details of the small source and the temporal structure of the x-ray source.

Changes to the manuscript:

New figure 1e:

270 ply by use of a small target. special target. The fact that
 271 measured source distributions along the horizontal direction
 272 (perpendicular to the needle axis) are even smaller than the
 273 needle itself, may be attributed to several factors. First, in
 274 the horizontal, laser radiation pressure can induce a con-
 275 verging plasma movement from the target surface to the
 276 center. This motion occurs roughly at the hole-boring ve-
 277 locity $v_h = 2a_0 c(Z/A \cdot m_e/m_p \cdot n_c/n_e)^{0.5}$, where Z is the
 278 average charge state of the ions, A ion mass number, m_e
 279 is the electron mass, m_p is the proton mass, n_c is the critical
 280 density and n_e is the electron density. Integration of v_h
 281 from before the pulse up to the peak intensity estimates a

282 movement for of the initial target surface between 250 nm
 283 ($Z = 10$, $n_e/n_c=300$) and 900 nm ($Z = 60$, $n_e/n_c=100$).
 284 While this hole boring motion is driven directly by the laser,
 285 another factor is the plasma expansion into vacuum, which
 286 also occurs at sides not facing the laser directly. This expan-
 287 sion reduces the density at the initial target-vacuum inter-
 288 face, and effectively leaves a reduced size of the high-density
 289 plasma region, where x-ray generation via Bremsstrahlung
 290 is strongest. The resulting overall reduction in bulk target
 291 size is usually observed when a high-power laser interacts
 292 with a curved surface, eg. simulations in references^{40,41}.
 293 Here, high-resolution 2D3V particle-in-cell simulations (fig.
 294 1e and methods) have been performed; they show that the 5
 295 μm target is transformed to a 3 μm plateau structure close
 296 to the original density when looked at through a central line-
 297 out along the laser propagation direction, i.e., representing
 298 the source distribution towards the sideport. The laser irra-
 299 diated side of this plateau structure shows an even smaller
 300 density-enhanced region due to the hole boring into the tar-
 301 get. Meanwhile, density around the plateau decreases expo-
 302 nentially. Note that generally 3D effects would be expected
 303 to increase the effect of target expansion on the effective
 304 source size. Further studies will elucidate the origin and
 305 temporal structure of the x-ray source to optimize this po-
 306 tentially useful feature.

Referee comment 17: “This projection is used to record...”

What is the effective magnification for the x-ray imaging shown in figure 2?

Our response: The magnification was given only indirectly by the geometry specified in the manuscript. It is about 2.5-fold. We added this information to the manuscript.

Referee comment 18: “Increasing magnification for the x-ray image in the point-projection...”

This phase-enhanced data is really stunning and I think you should promote to the main manuscript instead of supplementary material, as it really hammers down the impact of working with these needle targets and is a great complimentary imaging mode on top of the multi-species imaging. However, you

really need to explain the origin of these detector limited micron-scale source sizes to strengthen the portfolio of work shown here and validate your observations.

Our response: The tentative explanation above (comment 16) does give a reasonable idea as to why the source is small. Further investigations will be needed to completely understand its details, e.g. scaling with laser and target parameters. We moved the figure into the manuscript.

Referee comment 19: "...and produced radiographic contrast for...."

Selling yourself short here in this summary statement. Consider adding: "and have been applied for the first time to biological and technological samples." to highlight the novelty of the work presented.

Our response: The sentence has been changed accordingly.

Referee comment 20: "...facilitate high-resolution radiographies."

Can you achieve the ~ 5 micron source-limited imaging resolution AND multi-modal imaging simultaneously? What other factors does one need to consider for exploitation of this technique? How far back does the IP need to be so that you are no longer detector resolution limited and are taking advantage of tiny source size for high resolution? And then is there still enough photon density for single shot acquisition? Is point-projection proton radiography possible? So that you can have the sample close to the source and detectors far enough away to achieve high resolution. And then if the sample is close to the source point is it destroyed by intense radiation exposure? My point being: careful not to oversell here and make sure you manage expectations of laser-driven imaging technology. Remember that it won't be just laser-plasma people reading this paper, so you need to be explicit with the potential but also honest with limitations of the source.

Our response: This has indeed been tried to some extent (cf. figure 3). We simply placed the sample closer to the source. Meanwhile, we left the detector geometry for protons and x-rays the same as before. This is because the detector saturation of the CR39 (proton-) detector dictates the minimum detector distance from the source. I.e., in this specific imaging mode (protons + phase contrast), the sample had a distance $\gg 0$ cm from the CR39 detector. Since detector geometry is unchanged, this is still possible in single shot mode.

Figure 3: Proton and x-ray images with PCI contribution Left: photograph of an etched CR39 image recorded in the same shot as a phase contrast enhanced image. Right: X-ray image showing phase contrast contribution. Magnification in the proton image ca. 5 times, X-ray image ca. 12.3 times.

However, in the few laser shots tested in this configuration, we could not yet record fully satisfying images – another proton detector, e.g. a scintillating screen with different saturation properties may be better suited. For this reason, we preferred to leave this unfinished project out of the current manuscript.

Referee comment 21: "...could be extended to include neutron radiography in a single shot."

How are you proposing the neutron emission is generated? If via a mechanism that utilises the ion emission then this negates the possibility for simultaneous proton radiography? I can't see how proton-x-ray-neutron radiography is a compatible extension of this work. Unless the neutrons are from in-target reactions and the protons are from surface coulomb explosion as in present work? But I would love to be convinced otherwise if you can provide a brief extended discussion to this statement for the manuscript. Neutron-xray radiography on the other hand is absolutely compatible (if you can get your 2 keV x-rays through the neutron convertor material or if you shoot a deuterated needle target perhaps?) and is a very complimentary imaging modality.

Our response: We did some initial tests to deposit material onto the needles via evaporating superglue in an enclosed volume together with the needles. Results are not complete enough to make claims yet. If this technique was to be successful, there are possibilities to add deuterated layers for neutron generation from in-target reactions (e.g. Coulomb exploding protons from the initial target surface, penetrating through that deuterated layer). In the same sense as comment 20, since we have only started to explore these options, we removed the sentence from the current manuscript.

Referee comment 22: "small footprint"

Typo :)

Our response: Corrected.

Referee comment 23: "With such ideas, advantages of laser-driven sources could be enriched beyond their..."

This is a great statement summarising the unique strengths of laser-driven sources! Very strong justification for the current work. You should promote this sentence to much earlier in the manuscript and put it the abstract also. Maybe even second or third sentence of the abstract.

Our response: Done, hopefully to your satisfaction.

Referee comment 24: "...repetition rate are readily available."

And 5 - 10 Hz systems are currently coming on line too.

Our response: We agree and extended this outlook with higher rep-rate systems.

Referee comment 25: "The linear laser polarization..."

So it's parallel with the needle surface and therefore S polarisation? Or does the needle have a tapered edge/fast gradient taper so that the polarisation isn't exactly parallel? Or in the area of the laser spot can we assume the needle acts like a vertical wire and therefore laser E-field is parallel to surface?

Our response: Within the spot-size it is fair to treat the needle as a wire and the E-field as parallel. The needle-taper is easily slow enough for that (few um over 10s of um).

Referee comment 26: "Towards the sides, they explode..."

So in the region of interest and where you saw peaked spectrum the ion acceleration at play here is Coulomb explosion acceleration? Can you perform simulations to confirm your discussion here and validate your interpretation of the physics of this laser-needle interaction? This really feels like it is missing from the paper. Also, I think that this part should be promoted to the discussion section of the main manuscript and not hidden in methods as it explains the key physics of these interesting, and necessary for applications, spectra.

Our response: We admit that the exact proton spectrum is not easily accessible via numerical calculations or analysis for heterogenous targets. However, the basic physics behind the peaked proton spectrum via multispecies effects are known from earlier publications, e.g.,

- H. Schwoerer et al., Laser-plasma acceleration of quasi-monoenergetic protons from microstructured targets. *Nature*, 439(7075):445–448, Jan. 2006.
- A. Huebl et al., Spectral Control via Multi-Species Effects in PW-Class Laser-Ion Acceleration, <https://arxiv.org/pdf/1903.06428.pdf>
- Popov et al. A detailed study of collisionless explosion of single- and two-ion-species spherical nanoplasmals, PoP 2010, <https://dx.doi.org/10.1063/1.3474970>

The energy range and emission geometry is reasonable in context with other recent experiments at similar intensity (although with mono-species target, thus lacking the peaked spectral feature).

- L. Obst, et al., Efficient laser-driven proton acceleration from cylindrical and planar cryogenic hydrogen jets. *Scientific Reports*, 7(1):10248, 2017.

We added references and moved the explanation to the main text.

As a qualitative estimate, we performed additional electro-magnetic, fully-relativistic particle-in-cell simulations with PIconGPU. We would like to note that properly resolved kinetic simulations for high-density targets are even today very challenging and we invested 20 million NERSC core-hours to run 2D3V simulations with adequate resolution. Consequently, 3D3V simulations which could provide reliable, qualitative predictions for generated ion spectra and acceleration dynamics are not feasible for the full density of the tungsten target even on currently available, leadership-scale supercomputers. Hence, our qualitative comparison has to rely on reduced geometry (2D3V, both s- and p- polarizations) and a slightly reduced degree of ionization of the target. This approach generally underestimates target heating for s-polarization and over-estimates accelerated electron-density, especially at the target rear, due to geometric modeling of a laser line-focus along the third spatial dimension. Nevertheless, since the overall target geometry in-focus is otherwise cylinder-symmetric these simulations can still provide a qualitative estimate of the target compression and ion acceleration. These simulations (new fig. 1e, inset) do actually reveal a modulated spectral distribution and confirm our reasoning regarding its formation through space-charge effects; as expected from 2D simulations, the absolute energy scale is overestimated.

Changes to the manuscript:

New figure 1e:

83 Here, we present and characterize a novel laser-driven
 84 source, based on a micro-needle-target, that solves these
 85 issues in a combined approach. The **spectral** **spectral** char-
 86 acteristics of protons emitted towards the side is found to
 87 be peaked around 10 MeV, featuring a 20%-level energy
 88 spread (FWHM). **This benefits the** The protons stem from
 89 a nm-thin CH contaminant layer present on the Tungsten
 90 surface; the limited spatial extent translates to a limited
 91 spectral bandwidth observed towards the side, as the pro-
 92 tons explode away from the positively charged target into
 93 vacuum.

234 zontal plane. 2D3V particle-in-cell simulations (fig. 1e, de-
235 tails in methods and supplementary material IV) confirm the
236 qualitative shape of this spectral distribution; note that the
237 2D geometry is known to overestimate the absolute energy
238 scale. A proton source with a peaked energy spectrum
239 beyond 10 MeV has long been desired from a laser driven
240 accelerator with significant progress only compact laser driven ac-
241 celerators, because it would boost their relevance for many
242 applications originally developed with conventional accel-
243 erators, which per default emit a very narrow proton energy
244 spectrum. Possibly the most impactful application for a
245 tightly peaked proton spectrum, is the radiotherapy of can-
246 cer, where the sharp dose deposition at the Bragg peak
247 is used for targeted dose delivery to tumors. Significant
248 progress towards peaked spectra beyond 10 MeV have only
249 been reported in recent efforts exploring advanced accel-
250 eration mechanisms³⁵. In the present setup, the formation of
251 a peaked spectrum is facilitated by a combination of space-
252 charge effects between different ion species (highly ionized
253 tungsten ions and protons) and the localization of protons
254 in a very thin (nm-scale) contamination layer on a needle-
255 target that does not exceed the focus in size; both effects are
256 known to facilitate quasi-monoenergetic ion spectra³⁶⁻³⁸.

Referee comment 27:

“...magnification of 17.5-27.....”

I take it this was the magnification just for the phase contrast/high resolution shots? The multi-modal shots were with far lower magnification as the IP was only 0.5 m from the CR39. How far back was the IP for this high magnification projection imaging? And you still managed to get phase contrast imaging in a single shot? What is the divergence angle of the x-ray emission? Did it show any collimation?

Our response: 17.5-27 fold is the magnification used in source size measurements. The multimodal shots presented in the original submission's Figure 2 were at lower magnification (2.5 times).

But the source itself and detectors (CR39 and IP) were always placed at the *same* positions, also for the phase contrast imaging.

Magnification was varied only by moving the sample closer to the source (i.e., further away from CR39 and IP). Therefore, we did record these phase contrast enhanced images in single shots. Within our angles of observation we did not observe collimation effects.

Reviewer #2 (Remarks to the Author):

The major claim in the paper is a simultaneous single-shot radiographic imaging technique with a laser-driven x-ray and proton micro-source.

The proposal to use both x-rays and protons in a simultaneous radiographic imaging technique is not novel. The idea was reported by Nishiuchi et al., Journal of Physics: Conference Series 112, 042036 (2008). The novelty in the reported work concerns the micro-needle target that is used, which solves the outstanding problem of a simultaneous small virtual source for protons and a small x-ray spot size.

The paper will be of interest to others in the field and shows the value of this target interaction for bimodal radiographic applications. The paper will influence thinking in the field through the development and optimization of novel target interactions to engineer multi-modal and ultrafast radiographic techniques. The claims are convincing and the characterization techniques used appear sound.

The result would be more impactful with a dynamic demonstration to highlight the ultrafast nature of the bimodal radiographic technique. This aspect is not tested in the reported work, but represents an important part of the significant promise of ultrafast, laser-based sources (bimodal or otherwise). Such tests may be challenging, but feasible by splitting the short pulse beam to generate a driver to excite a dynamic radiographic object.

The claims are appropriately discussed in the context of previous literature. The manuscript is clearly written and they have not oversold their claims. They have been fair in the treatment of previous work and sufficient detail has been provided that the work could be reproduced.

Our response: We thank the referee for this very positive evaluation of our work. We agree that the proposed imaging of dynamic processes represents a challenging and high-impact next step for this method. The lead author of this study currently works on the development of a laser-system with two independently tunable high-power laser-pulses, suitable for this kind of experiments (and more generally for pump-probe experiments).

Meanwhile, the imaging of biological and technological samples represents the first real demonstration of such a method, and besides relevance for fast imaging, it allows our community to engage with researchers of other disciplines that become interested in actually using the source. We consider this an important step to advance our field.

Reviewer #3 (Remarks to the Author):

The paper “Simultaneous single-shot radiographic imaging with a laser-driven x-ray and proton micro-source” by Ostermayr et al., describes the use of a single intense laser pulse to generate sources of protons and x-rays for radiography. The experimenters use a low repetition rate high energy laser incident onto a tungsten microneedle target. This produces an interesting proton source, emitted at all angles as well as an x-ray source also emitted at all angles due to bremsstrahlung or possibly atomic emission. There is no theoretical discussion and no modelling of the interaction.

The paper involves the use of these radiation sources for radiography of a biological object simultaneously with both x-rays and protons.

Positive aspects of this work are:

1) The authors measure an unusual peaked proton spectrum in the transverse direction, although this is unexplained.

Our response: A qualitative explanation was given for this spectrum, which is also known from literature, i.e. space-charge effects and the spatial limitation of protons on the target cause a modulated spectrum. The cylindrical geometry should (and does) not have a significant impact on these very basic underlying physical processes.

We admit that the exact proton spectrum is not easily accessible via numerical calculations or analysis for heterogeneous targets. However, the basic physics behind the peaked proton spectrum via multispecies effects are known from earlier publications, e.g.,

- H. Schwoerer et al., Laser-plasma acceleration of quasi-monoenergetic protons from microstructured targets. *Nature*, 439(7075):445–448, Jan. 2006.
- A. Huebl et al., Spectral Control via Multi-Species Effects in PW-Class Laser-Ion Acceleration, <https://arxiv.org/pdf/1903.06428.pdf>
- Popov et al. A detailed study of collisionless explosion of single- and two-ion-species spherical nanoplasmas, *PoP* 2010, <https://dx.doi.org/10.1063/1.3474970>

The energy range and emission geometry is reasonable in context with other recent experiments at similar intensity (although with mono-species target, thus lacking the peaked spectral feature).

- L. Obst, et al., Efficient laser-driven proton acceleration from cylindrical and planar cryogenic hydrogen jets. *Scientific Reports*, 7(1):10248, 2017.

We added references and moved the explanation to the main text.

As a qualitative estimate, we performed additional electro-magnetic, fully-relativistic particle-in-cell simulations with PIConGPU. We would like to note that properly resolved kinetic simulations for high-density targets are even today very challenging and we invested 20 million NERSC core-hours to run 2D3V simulations with adequate resolution. Consequently, 3D3V simulations which could provide reliable, qualitative predictions for generated ion spectra and acceleration dynamics are not feasible for the full density of the tungsten target even on currently available, leadership-scale supercomputers. Hence, our qualitative comparison has to rely on reduced geometry (2D3V, both s- and p- polarizations) and a slightly reduced degree of ionization of the target. This approach generally underestimates target heating for s-polarization and over-estimates accelerated electron-density, especially at the target rear, due to geometric modeling of a laser line-focus along the third spatial dimension. Nevertheless, since the overall target geometry in-focus is otherwise cylinder-symmetric these simulations can still provide a qualitative estimate of the target compression and ion acceleration.

These simulations (new fig. 1e, inset) do actually reveal a modulated spectral distribution and confirm our reasoning regarding its formation through space-charge effects; as expected from 2D simulations, the absolute energy scale is overestimated.

83 Here, we present and characterize a novel laser-driven
 84 source, based on a micro-needle-target, that solves these
 85 issues in a combined approach. The **spectral** character-
 86 istics of protons emitted towards the side is found to
 87 be peaked around 10 MeV, featuring a 20%-level energy
 88 spread (FWHM). **This benefits the** The protons stem from
 89 a nm-thin CH contaminant layer present on the Tungsten
 90 surface; the limited spatial extent translates to a limited
 91 spectral bandwidth observed towards the side, as the pro-
 92 tons explode away from the positively charged target into
 93 vacuum.

234 zontal plane. 2D3V particle-in-cell simulations (fig. 1e, de-
 235 tails in methods and supplementary material IV) confirm the
 236 qualitative shape of this spectral distribution; note that the
 237 2D geometry is known to overestimate the absolute energy
 238 scale. A proton source with a peaked energy spectrum
 239 beyond 10 MeV has long been desired from a **laser driven**
 240 **accelerator with significant progress only compact laser driven**
 241 **accelerators**, because it would boost their relevance for many
 242 applications originally developed with conventional accel-
 243 erators, which per default emit a very narrow proton energy
 244 spectrum. Possibly the most impactful application for a
 245 tightly peaked proton spectrum, is the radiotherapy of cancer,
 246 where the sharp dose deposition at the Bragg peak
 247 is used for targeted dose delivery to tumors. Significant
 248 progress towards peaked spectra beyond 10 MeV have only
 249 been reported in recent efforts exploring advanced accel-
 250 eration mechanisms³⁵. In the present setup, the formation of
 251 a peaked spectrum is facilitated by a combination of space-
 252 charge effects between different ion species (highly ionized
 253 tungsten ions and protons) and the localization of protons
 254 in a very thin (nm-scale) contamination layer on a needle-
 255 target that does not exceed the focus in size; both effects are
 256 known to facilitate quasi-monoenergetic ion spectra³⁶⁻³⁸.

2) An unusually small source size of x-rays in transverse direction is measured. There is both bremsstrahlung and atomic emission from such interactions which would have different directional, source size and pulse duration characteristics. There was not much data on source size variation with laser properties and target properties presented in the paper.

Our response:

Spectrum: More information regarding x-ray generation can be obtained in experiments with direct spectral resolution (in contrast to retrieval algorithms based on a specific distribution as used in our paper). Our collaborator Dr. Paul Neumayer (GSI Darmstadt) kindly shared data from shots with the PHELIX laser (up to 180 J, 5 μm FWHM focus diameter, 500 fs FWHM duration, 5 μm focus) onto 5 μm thick tungsten wires. This is a fairly comparable scenario to ours. X-ray spectra were recorded with a HOPG spectrometer.

Figure 3: (top) Low energy (mJ) shot on tungsten wire at PHELIX laser. (bottom) High energy (180 J) shot on tungsten wire at PHELIX showing Bremsstrahlung continuum plus broad lines. Note: all scales are linear. Figure by courtesy of Dr. Paul Neumayer.

A low energy shot (1 mJ) shows almost exclusively the tungsten L alpha and L beta lines. At full energy of 180 J, the spectrum consists of a continuum to which a temperature of ca. 3.5 keV can be fitted, and some smeared out structures on top of that, which have been identified as transitions into the L-shell of highly ionized tungsten. This explains why our simple Bremsstrahlung fit performs reasonably well. A short version of the above discussion is also part of

- [1] L. Antonelli et al., X-ray phase-contrast imaging for laser-induced shock waves. EPL (Europhysics Letters), 125(3):35002, Feb 2019.

We added this work as citation to the manuscript.

183 tion factor. With a fit temperature of $k_B T_e = 2360 \pm 140$
 184 $k_B T_e = 8060 \pm 940$ eV (95% confidence level) and con-
 185 sidering emission in the full solid angle, this spectrum
 186 contains an energy of 8.9 1.6 mJ, of which more than 3.8
 187 1.2 mJ are emitted at energies higher than 2 keV. It shall
 188 be mentioned that recent experiments at comparable con-
 189 ditions using tungsten wire targets^{33,34} found similar spectra,
 190 with high-resolution HOPG spectrometers additionally iden-
 191 tifying broadened emission structures around 8.2-8.4 keV
 192 from electron transitions into the L-shell of highly ionized
 193 tungsten. However, the Bremsstrahlung content was iden-
 194 tified as dominant, consistent with our fit. Note that our
 195 detector setup is laid out for these relatively low-energy x-
 196 rays. Higher energy MeV-scale x-rays may also be emitted
 197 from the laser-plasma interaction, but are not expected to
 198 significantly contribute to the recorded signal.

Source Size: We suggest an intuitive idea as to why the x-ray source size is small. Targets with curved surfaces are known to facilitate converging movements that can compress the target. The temporally integrated source size will be dominated by the time of highest temperature in the largest number of bulk electrons.

For the converging movement of the bulk, we can estimate its relevance via the hole boring velocity $v_b = 2 a_0 c (Z/A * m_e/m_p * n_c/n_e)^{0.5}$, where Z is the degree of ionization, A ion mass number, m_e is the electron mass, m_p is the proton mass, n_c is the critical density and n_e is the electron density. Integration of v_b from before the pulse to the peak intensity yields a movement for of the initial target surface of

- 250 nm with parameters: $Z=10, n_e/n_c=300$
- 900 nm with parameters: $Z=60, n_e/n_c=100$.

Another contributing factor is the plasma expansion into vacuum. This expansion reduces the density at the initial target-vacuum interface, and effectively leaves a reduced size of the high-density plasma region, where x-ray generation via Bremsstrahlung is expected to be strongest.

2D particle-in-cell simulations (PConGPU, new fig. 1e main figure) shows agreement with the ideas outlined above. We observed both the effect of hole boring and plasma expansion, leading to an overdense region of just 3 μm diameter with an even smaller density enhanced region caused by laser hole boring into the target. Around this plateau, density drops exponentially.

274 achieved simply by use of a small target. special target. The
 275 fact that measured source distributions along the horizontal
 276 direction (perpendicular to the needle axis) are even smaller
 277 than the needle itself, may be attributed to several factors.
 278 First, in the horizontal, laser radiation pressure can induce
 279 a converging plasma movement from the target surface to
 280 the center. This motion occurs roughly at the hole-boring
 281 velocity $v_h = 2a_0c(Z/A \cdot m_e/m_p \cdot n_c/n_e)^{0.5}$, where Z is

282 the average charge state of the ions, A ion mass number,
 283 m_e is the electron mass, m_p is the proton mass, n_c is the
 284 critical density and n_e is the electron density. Integration of
 285 v_h from before the pulse up to the peak intensity estimates
 286 a movement for of the initial target surface between 250 nm
 287 ($Z = 10$, $n_e/n_c=300$) and 900 nm ($Z = 60$, $n_e/n_c=100$).
 288 While this hole boring motion is driven directly by the laser,
 289 another factor is the plasma expansion into vacuum, which
 290 also occurs at sides not facing the laser directly. This expansion
 291 reduces the density at the initial target-vacuum interface,
 292 and effectively leaves a reduced size of the high-density
 293 plasma region, where x-ray generation via Bremsstrahlung is
 294 strongest. The resulting overall reduction in bulk target size
 295 is usually observed when a high-power laser interacts with
 296 a curved surface, eg. simulations in references^{40,41}. Here,
 297 high-resolution 2D3V particle-in-cell simulations (fig. 1e,
 298 methods and supplementary material IV) have been performed;
 299 they show that the 5 μm target is transformed
 300 to a 3 μm plateau structure close to the original density
 301 when looked at through a central lineout along the laser
 302 propagation direction, i.e., representing the source distribution
 303 towards the sideport. The laser irradiated side of this
 304 plateau structure shows an even smaller density-enhanced
 305 region due to the hole boring into the target. Meanwhile,
 306 the density around the plateau decreases exponentially. Further
 307 studies will elucidate the origin and temporal structure
 308 of the x-ray source to optimize this potentially useful feature.
 309

We know that a single laser will seldomly suffice to study all the corners of a multidimensional parameter space. It is also not what we claim. This paper is focused primarily on a first of its kind application of one particular source (of which variations will exist).

Problematic aspects of this work are:

1) X-ray phase contrast imaging and proton imaging have been done previously using solid targets although separately. The justification for why one would want to do both at the same time in this geometry is not convincing.

Our response: No clear reason is given here that would disarm our arguments. To highlight those arguments in the paper, we moved the most important ones to the abstract

Novel imaging methods have revealed some of nature's most fascinating and unexpected secrets. Today, radiographic and Radiographic and tomographic imaging with x-rays and protons are an omnipresent tool in basic research and applications relevant to industry, material science, military and medical diagnostics. In some of these examples cases, the information contained in both modalities is can be valuable in principle, but difficult to access simultaneously. Laser-driven solid-density plasma-sources have long been known to deliver both kinds of radiation at the same time, but mostly single modalities have been explored for applications, aiming to replicate use-cases of conventional single-species sources. Their potential for bimodal bi-modal radiographic imaging has never been fully realized, due to conceptual problems in generating appropriate sources and recording well separated images. Here, we report on the generation of proton and x-ray sources micro-sources in a laser-plasma interaction of the focused Texas Petawatt laser with a solid-density, micrometer-sized Tungsten tungsten needle. We demonstrate their unique capabilities in terms of spectral and spatial distributions and use them for bimodal apply them for bi-modal radiographic imaging of biological and technological objects in a single laser shot. The small x-ray source additionally enables images with phase-contrast contribution. Bi-modal imaging could serve as a compact test environment for studies of the conversion process from x-ray attenuation images to proton stopping maps. The combination of both images via post-processing could use the higher spatial resolution of the x-ray image, which does not suffer from multiple Coulomb scattering, to improve the proton image. Such approaches can directly benefit from the single-shot bi-modal imaging, where both images are automatically registered on top of each other in time and space. The short pulse lengths and the fixed temporal relation between protons and x-rays could benefit recordings of moving objects, e.g., living/breathing organisms or plasma instabilities. With such ideas, advantages of laser-driven sources could be enriched beyond their small footprint by embracing their additional unique properties, including the spectral bandwidth, small source size and multi-mode emission.

Having both proton source and X-ray source combined in the very same target adds the certainty of their overlay to within micrometers, which is harder to achieve with two separated targets. Community feedback as well as Referees #1 and #2 indicate that the suggested applications are widely considered relevant.

2) It would obviously provide better and more controllable simultaneous radiography data by just splitting the Petawatt beam in two and having two optimized sources with temporal control. This could also done with any existing multibeam short pulse system (ARC, OMEGA, ORION etc.)

Our response:

There are clear benefits to both approaches – given our response above, it is not obvious that two individual sources are always the better solution, e.g., since the better controls come at the cost of experiment complexity. That said, the lead author currently works on the development of two individually tunable laser pulses for these kinds of experiments. If things were as simple as suggested by the referee, the proposed application would have been demonstrated by now.

3) The main scientific interest in proton radiography is in measuring time resolved fields, however here the protons are used to measure only the density providing the same information as the x-rays but with less resolution. For field measurements with a single beam as the source the x-ray and proton measurements would be separated in time by many picoseconds so this wouldn't be necessarily be useful.

Our response: Time resolved imaging of electro-magnetic fields is an exciting, but small, subset of applications for proton radiography. Depending on the system timescale in question and the spatial distance between source and object (can be sub-mm-scale, hence timing can be ps-range indeed), our technique might still be applicable. More importantly, we line out several other application scenarios in which our idea *could* be useful.

Or to say it with Arthur Schawlow: “We thought it [the maser/laser] might have some communications and scientific uses, but we had no application in mind. If we had, it might have hampered us and not worked out as well.”

4) Why is the scattered light only in the forward direction in the experiment? This suggests that the microneedle target is destroyed by the pre-pulse before the interaction and the interaction is with an underdense plasma.

Our response: The transmitted light can be explained by the compression of the needle (discussed earlier), which naturally causes more light to pass. If this plasma were to be underdense, its extent would have to be so enormous that a source size of few μm could never have been measured.

More significantly, an underdense plasma does not emit this kind of proton and x-ray source (if we trust the past few decades of simulations and experiments).

In fact, recently published results (L. Obst-Huebl et al.) concerning laser ion acceleration from shots at few- μm thick cylindrical targets are compatible with our observation of transmitted laser-light (>50%).

These results indicate that light-transmission can modulate the ion beam in the laser propagation direction through interactions with residual gas, which gives yet another reason to perform radiographies outside of the transmitted laser beam cone.

Obst-Huebl et al., All-optical structuring of laser-driven proton beam profiles, Nature Communications, <https://www.nature.com/articles/s41467-018-07756-z>

5) What is the spectrum of the x-rays? How are the components due to atomic emission and bremsstrahlung different with respect to source size, spectrum etc.

Our response: See above. This question is a repetition.

In conclusion this is an interesting paper however I don't believe that it is of sufficient importance to be published in Nature Communications. It is actually more appropriate for Review of Scientific Instruments or Applied Physics Letters. For a high impact journal such as Nature Communications the authors would need to use this source to make a new scientific measurement which couldn't have been done in another way. They also need to do a more in-depth investigation of the micro-needle interaction which would include numerical modeling.

Our response: We thank the referee for his final evaluation of our paper as "interesting". We think that our paper does present an otherwise undoable measurement: the single source, single shot simultaneous radiography has not been done before with a laser-plasma device. We hope that our extensive revision of this manuscript and the added information including numerical simulations will convince the referee to support the publication in Nature Communications.

Reviewers' comments:

Reviewer #1 (Remarks to the Author):

The authors have addressed the reviewer's comments effectively and have produced a much stronger manuscript, which I can now recommend for publication.

The PIC simulations reveal the physics at play and a well-justified interpretation is communicated in the manuscript.

Be mindful in future of gendered pronouns in response to reviewers. 'His' or 'her' can be replaced with 'their' to make the response appropriate for everyone.

Reviewer #3 (Remarks to the Author):

The authors of the paper "Simultaneous single-shot radiographic imaging using a laser-driven x-ray and proton micro-source" have addressed some of my previous comments in the new version of their paper. However fundamentally I do not believe this to be a result of sufficient importance for publication in Nature Communications.

This is much more suitable for an Applied Physics journal since there is no new physics described.

The most interesting application for simultaneous proton and x-ray imaging would be if field and density dynamics could be measured at the same time, however the time difference between the x-rays and the protons makes the proposed technique not particularly valuable. Presumably that is why the previous paper on simultaneous x-ray and proton probing was not followed up and was only published in an IFSA conference proceedings.

I suggest that the authors publish this work in an Applied Physics journal (where all new diagnostic techniques are generally described) and then perform an experiment in which this simultaneous measurement of proton and x-ray images using only one laser pulse provides some new information.

We thank the reviewer for their response. We note that concerns regarding applicability and relevance of our work had been addressed in previous communication. The review does not contain an argument to substantiate such concerns. In this latest review, we find no attempt to consider (or to argue) our previously delivered arguments regarding the wide-ranging and immediate impact of our work as a source, e.g., in biomedical imaging. In particular:

- Reviewer statement: Presumably that [not imaging dynamic plasma processes] is why the previous paper on simultaneous x-ray and proton probing was not followed up and was only published in an IFSA conference proceedings.

We believe this statement is wrong. As described in our paper, previously published attempts were indeed preliminary in that they were limited to binary (black and white) imaging of very thin (10s of μm) structures. They were also very constrained in particle energies due to limitations inherent to their experiment setup (e.g., no separation of laser/electron/proton/x-ray beams, no small source size at high laser intensities). This made their *source* much less viable and useful for many applications. Our novel source overcomes all these obstacles at once, with one elegant approach, allowing us to produce contrast resolved images with smaller source sizes. Arguing with the intrinsic limitations of those preliminary studies is simply not adequate for our approach.

- Reviewer statement: This is much more suitable for an Applied Physics journal since there is no new physics described. The most interesting application for simultaneous proton and x-ray imaging would be if field and density dynamics could be measured at the same time, however the time difference between the x-rays and the protons makes the proposed technique not particularly valuable. [...] I suggest that the authors publish this work in an Applied Physics journal (where all new diagnostic techniques are generally described) and then perform an experiment in which this simultaneous measurement of proton and x-ray images using only one laser pulse provides some new information.

We believe this statement is wrong. In its very core, our paper describes in much detail a *novel source* including its physics, its key parameters, its multiple application scenarios, and (on top of that) even demonstrates one such relevant example using its novel capabilities. In particular, our manuscript dives deep in measuring and explaining the special properties of this source, namely the narrow-band proton source, the keV x-ray source and very small x-ray and proton source sizes. All of these properties are of high-quality and present unprecedented results for solid density targets, which have not been observed before (and especially, not in combination). It is worth highlighting again, as presented in the manuscript, that such sources are valuable for many applications. Amongst others, surrounding single-shot bi-modal imaging, which are foreseen in biomedical imaging, industrial applications (e.g., advanced material testing), plasma imaging and pump-probe setups. Both referees 1 and 2 acknowledge the relevance and

novel source physics contained in our manuscript directly in their reports. There exists a long history of similar (single species) source papers published in high impact journals (e.g., Hegelich et al. 2006; Kneip et al. 2010; Rousse et al. 2004; Ta Phuoc et al. 2012). And for good reason: novel particle and light sources are the drivers of discovery science and applications in many fields.

This is not limited to plasma physics. The premise of this being the most promising application is very questionable, and judging any source only based on its relevance to plasma physics seems unjustified. To us, as a department of medical physics, the bimodal imaging of biological specimens, as reported here, appears far more important and promising than probing plasmas. Especially, bi-modal imaging can help to improve pre-clinical and clinical radiotherapy by combining the complementary strengths of both imaging modalities, with charged and neutral particles, in a single compact setup. The interdisciplinary nature of this work and its expected broader impact are precisely what makes it suitable for publication in Nature Communications. Following the reviewer's logic, most of *plasma physics* itself (oftentimes published in Nature journals) would have to be considered preliminary and not high impact, for not nurturing substantial applications to date. Therefore, we feel that, if anything, the application of our source to some plasma physics should be published in an Applied Physics Journal (a similar path has been pursued with other sources mentioned above).

Changes: It became clear from the reviewer's communication that certain points needed more context and clarification. In order to better communicate the central ideas and goals of this work, we made several changes to the manuscript.

Since this is essentially a source paper, with an application on top, we changed its title to **"Laser-driven x-ray and proton micro-source and application to simultaneous single-shot bi-modal radiographic imaging"**.

While there is no other *comparable* laser driven (or conventional) bi-modal source published in a similar context, we understand the need for comparisons to laser driven single species sources. We added this discussion. We also expanded the discussion of perspectives (in particular in the biomedical context) in the manuscript.

With these points we hope the reviewer can better recognize the important and new aspects of this paper and can appreciate its relevance for Nature Communications, as it clearly speaks to more than just the laser plasma physics community that may be the specialized audience of related Applied Physics journals.

References:

Hegelich, B. M., B. J. Albright, J. Cobble, K. Flippo, S. Letzring, M. Paffett, H. Ruhl, J. Schreiber, R. K. Schulze, and J. C. Fernández. 2006. "Laser Acceleration of Quasi-Monoenergetic MeV Ion Beams." *Nature* 439 (7075): 441–44.

Kneip, S., C. McGuffey, J. L. Martins, S. F. Martins, C. Bellei, V. Chvykov, F. Dollar, et al. 2010. "Bright Spatially Coherent Synchrotron X-Rays from a Table-Top Source." *Nature Physics* 6 (12): 980–83.

Rousse, Antoine, Kim Ta Phuoc, Rahul Shah, Alexander Pukhov, Eric Lefebvre, Victor Malka, Sergey Kiselev, et al. 2004. "Production of a keV X-Ray Beam from Synchrotron Radiation in Relativistic Laser-Plasma Interaction." *Physical Review Letters* 93 (13): 135005.

Ta Phuoc, K., S. Corde, C. Thaury, V. Malka, A. Tafzi, J. P. Goddet, R. C. Shah, S. Sebban, and A. Rousse. 2012. "All-Optical Compton Gamma-Ray Source." *Nature Photonics* 6 (5): 308–11.